



# Age distribution, extractability, and stability of mineral-bound organic carbon in central European soils

Marion Schrumpf[1], Klaus Kaiser[2], Allegra Mayer[3], Günter Hempel[4], Susan Trumbore[1]

[1]Max-Planck-Institute for Biogeochemistry, Jena, D-07745, Germany
[2]Soil Science and Soil Protection, Martin Luther University Halle-Wittenberg, Halle (Saale), 06120, Germany
[3]Department of Environmental Science, Policy and Management, University of California, Berkeley, Berkeley, CA 94720, USA
[4]Institute of Physics, Martin Luther University Halle-Wittenberg, Halle (Saale), 06120, Germany

*Correspondence to*: Marion Schrumpf (mschrumpf@bgc-jena.mpg.de)

**Abstract.** The largest share of total soil organic carbon (OC) is associated with minerals. The portions and turnover of stable and faster cycling mineral-associated carbon (MOC) as well as the determining factors across different soils and soil depths are still unknown. Bioavailability of MOC is supposedly regulated by desorption but instead, its stability was so far mostly tested by exposure to chemical oxidation. Therefore, we determined the extractability of MOC into a mixture of 0.1 M NaOH and 0.4 M NaF as a measure for maximal potential desorbability, and compared it with maximal potential oxidation

in heated $H_2O_2$. We selected samples of three soil depth increments (0-5 cm, 10-20 cm, 30-40 cm) of five typical soils of the mid-latitudes, differing contents of clay and pedogenic oxides, and being under different land use. Extracts and residues were analyzed for OC and [14]C contents, and further chemically characterized by CPMAS-[13]C-NMR. We hypothesized NaF-NaOH extraction to remove less and younger MOC than $H_2O_2$ oxidation, and extractable MOC to be less and relatively older in subsoils and soils with high contents of pedogenic oxides.

A surprisingly constant portion of 58±11% (standard deviation) of MOC was extractable across soils, independent of depths, mineral assemblage, or land use. NMR spectra revealed strong similarities of the extracted organic matter, with more than 80% of OC in the O/N alkyl and alkyl C region. Total MOC amounts were linked to the content of pedogenic oxides across sites, independent of variations in total clay. The uniform MOC desorption could therefore be the result of pedogenic oxides dominating the overall response of MOC to extraction.

While bulk MO[14]C values suggested differences in OC turnover between sites, these were not linked to differences in MOC extractability. As expected, OC contents of residues had smaller [14]C contents than extracts, suggesting that non-extractable OC is older. However, [14]C contents of extracts and residues were strongly correlated and proportional to bulk MO[14]C, but not dependent on mineralogy. Also along soil profiles, where increasing MOC ages indicate slower turnover with depth, neither MOC extractability nor differences in [14]C between extracts and residues changed. Increasing bonding strength with

soil depths did therefore not cause the [14]C depth gradients in the studied soils.

Although $H_2O_2$ removed 90±8% of the MOC, the [14]C content of the OC removed was similar to that of the NaF-NaOH-extracted OC, while oxidation residues were much more [14]C-depleted. Different chemical treatments apparently remove OC



of the same continuum, leaving increasingly older residues behind the more OC being removed. Different from the extractions, higher contents of pedogenic oxides seemingly slightly increased the oxidation-resistance of MOC, but this

higher $H_2O_2$-resistance did not coincide with older MOC or oxidation residues.

Our results indicate that total MOC was dominated by OC interactions with pedogenic oxides rather than clay minerals, so that no difference in MOC extraction in NaF/NaOH, and thus, bond type or strength between clay-rich and poor sites was detectable. This suggests that site-specific differences in $MO^{14}C$ and their depth declines are driven by the accumulation and exchange rates of OC at mineral surfaces. Accordingly, future research on $M^{14}OC$ should focus on soil and ecosystem

properties driving dissolved organic matter formation, composition and transport along soil profiles.

## 1 Introduction

The persistence of organic matter (OM) in soil is one key control of atmospheric $CO_2$ concentrations. Binding to minerals is considered an effective pathway of stabilizing otherwise degradable OM against microbial mineralization (Schmidt et al., 2011; Lehmann and Kleber, 2015; Hemingway et al., 2019) and in many soils most of the contained organic carbon (OC) is

bound to minerals (Kleber et al., 2015; Kögel-Knabner et al., 2008; Cotrufo et al., 2019). Nevertheless, we still lack fundamental knowledge on the drivers of the stability and turnover of mineral-bound OC (MOC) along soil profiles and across sites.

The formation of MOC involves sorption to reactive minerals such as phyllosilicate clays and pedogenic aluminum (Al) and iron (Fe) oxi-hydroxides (Schrumpf et al., 2013; Kögel-Knabner et al., 2008; Kaiser and Guggenberger, 2000; Khomo et al.,

2017). Along with plant-derived decomposition products, microbial residues and metabolites are important precursors and sorbates for MOC formation (Avneri-Katz et al., 2017; Chenu and Stotzky, 2002; Cotrufo et al., 2015; Kalbitz et al., 2005; Kallenbach et al., 2016). Experimental studies showed that sorption to minerals reduces OC mineralization (e.g. Kalbitz et al., 2005; Jones and Edwards, 1998; Eusterhues et al., 2014; Porras et al., 2018). Radiocarbon ($^{14}C$) analyses confirmed the greater stability of MOC in soils, showing that it typically is of older average ages than OC not bound to minerals (Schrumpf

et al., 2013; Kögel-Knabner et al., 2008; Hemingway et al., 2019; Heckman et al., 2018). Several field and incubation studies suggested that total amounts as well as stability of MOC depend on the soil mineral composition and increases with the amount of pedogenic Al and Fe oxi-hydroxides (Bruun et al., 2010; Torn et al., 1997; Porras et al., 2017; Rasmussen et al., 2006). Other studies found only correlations between Al and Fe oxi-hydroxides and MOC concentrations but not with the $^{14}C$ content of MOC and accordingly its average age (Herold et al., 2014; Khomo et al., 2017; Schrumpf et al., 2013).

Studying MOC turnover and its drivers in soil is complicated because, similar to bulk soil OC, it is a mixture of young, less well stabilized, and older, presumably better stabilized, carbon (Trumbore et al., 1989; Swanston et al., 2005; Schrumpf and Kaiser, 2015; Koarashi et al., 2012). Various chemical methods, such as acid hydrolysis and chemical oxidation, have often been applied to distinguish faster and slower cycling fractions (e.g. Mikutta et al., 2006; Jagadamma et al., 2010; Helfrich et al., 2007; Six et al., 2002; Paul et al., 2001; Eusterhues et al., 2003). They all separated bulk soil OC or MOC into younger



and older fractions but differed in the extent of OC removal and the $^{14}$C contents of obtained residues. The oxidants $H_2O_2$ or $Na_2S_2O_8$ were more effective in removing OC from samples than NaOCl, and their residues were older (Jagadamma et al., 2010; Helfrich et al., 2007). With increasing presence of Al and Fe oxi-hydroxides in soils, larger amounts of OC resisted chemical oxidation, suggesting that binding to those minerals provides some protection against oxidative attack (Mikutta et al., 2006; Kleber et al., 2005; Eusterhues et al., 2005). However, there are indications that such chemically defined fractions

are possibly not causally related to MOC bioavailability or persistence in soils (Poirier et al., 2003; Mikutta and Kaiser, 2011; Helfrich et al., 2007; Jagadamma et al., 2010; Paul et al., 2008).

Release of mineral-bonded OC, either by desorption or upon mineral dissolution under changing environmental conditions, can support or even be prerequisite for its microbial degradation (Keil et al., 1994; Mikutta et al., 2007). Desorption of OM under conditions similar to those during the formation of MOC was typically small (e.g. Gu et al., 1994), but increased in the

presence of competing ions such as $SO_4^{2-}$ or $H_2PO_4^-$, and was largest when solution pH was raised (e.g. Kaiser and Zech, 1999; Kaiser and Guggenberger, 2007). While some studies observed that the release of OC from minerals into alkaline solutions increased with the minerals' OC loading (Kaiser and Guggenberger, 2007, Kaiser et al., 2007), others showed larger OC desorption in subsoils despite smaller OC concentrations (Kaiser and Zech, 1999; Mikutta et al., 2009). Larger desorption and biodegradation of OM bonded to phyllosilicates than of OM bonded to Al and Fe oxi-hydroxides has been

attributed to differences in bonding strength. The sorption of OM to Al and Fe oxi-hydroxides involves surface complexation, which results in strong chemical bonds, while the sorption between OM and phyllosilicates is largely due to weaker cation bridges and van der Waals bonds (Singh et al., 2016; Mikutta et al., 2007). Presence and amount of Al and Fe oxi-hydroxides, therefore, typically decreased both, desorption, and mineralization rates (Oren and Chefetz, 2012; Saidy et al., 2012; Singh et al., 2017). Unfortunately, OC desorption was so far mostly studied using model minerals in laboratory

experiments, and observation times for desorption or mineralization were short relative to carbon residence times in soil. Mikutta et al. (2010) analyzed the $^{14}$C contents of MOC after removing all potentially desorbable OC by extracting soil with a combination of NaF and NaOH. This extraction allows for studying the potential displacement of complexed organic functionalities by competing $OH^-$ and $F^-$ anions, and the rise in pH (Kaiser et al., 2007). Consistently younger OM was extracted from MOC of topsoils, supporting the idea that desorbable OC turns over faster than more strongly bond OC.

Results for subsoils were more variable (Mikutta et al., 2010). Along with $^{14}$C contents, the chemical composition of OC extractable into alkaline solutions also changes with soil depth, with subsoils containing less lignin-derived aromatics but more O-alkyl C, possibly of microbial origin (Möller et al., 2000; Mikutta et al., 2009). Desorption of OC from MOC has, to our knowledge, so far not been studied systematically across soil types. Assuming that potential OC desorption is closer to the mechanism behind mineral protection of soil OC than the abovementioned oxidative or hydrolytic extractions, factors

controlling amount, age, and composition of desorbable OC in soils deserve attention.

In order to test if maximum desorption is a suitable indicator for the labile portion of MOC, we took advantage of a former experiment, where MOC was isolated from soils by density fractionation (heavy fraction at a density cutoff of >1.6 g cm$^{-3}$, HF) for a range of sites across Europe (Schrumpf et al., 2013). We selected five sites to have a range of the amount and type



of clays and pedogenic oxides as well as of land use, and, accordingly, amount and quality of litter input. Samples from three

soil depths were extracted with a combination of NaOH and NaF, and analyzed for amount, composition, and $^{14}$C contents of extracted OC. Results are compared to the amount and age of OC oxidizable by heated $H_2O_2$, which was shown to remove the largest and isolate the oldest part of MOC (Helfrich et al., 2007; Jagadamma et al., 2010). The desorption experiment is supposed to addresses mineral protection as stabilization mechanism, while the oxidation treatment should rather address the chemical recalcitrance of MOC.

We hypothesize that:

(1) Extraction in NaF/NaOH releases a potentially desorbable, weaker bound fraction from total MOC, which is younger than the stronger bound, probably better stabilized residue fraction. Accordingly, total MOC should be younger with larger portions of total MOC being extractable.

(2) The portion of total MOC extractable by NaF/NaOH decreases with increasing contents of pedogenic oxides, which

form strong bonds with OM.

(3) The portion of young NaF/NaOH-extractable MOC declines from topsoils to subsoils with declining OC loading and increasing stability and age of MOC.

(4) The strong oxidizing agent $H_2O_2$ removes more of the total MOC than NaF/NaOH. Assuming that OM becomes increasingly older the stronger it is bound to minerals, both oxidizable and non-oxidizable OC should be older than

the extractable and non-extractable OC fractions.

(5) Extractability of mineral-bound OM with NaF/NaOH and oxidation of OC are related to the chemical composition of sorbed OC, and thus, vary with land use and soil depth. In particular, MOC should be less prone to desorption and oxidation, and accordingly older, where organic acids capable to from strong bonds with minerals prevail.

## 2 Materials and Methods

Composition and age structure of MOC were studied on heavy fraction (HF) material obtained at five of the sites presented by Schrumpf et al. (2013). The sites include two deciduous forests developed on loess at Hesse (France, Cambisol) and loess over limestone at Hainich (Germany, Cambisol). The grassland site Laqueuille (France, Andosol) and the coniferous site Wetzstein (Germany, Podzol) are characterized by large contents of pedogenic oxides. The fifth site, a cropland at Gebesee (Germany, Chernozem), reveals a plow layer down to 30 cm, and large contributions of old OC throughout the profile

(Schrumpf et al., 2013). The MOC fraction was separated using two-step sequential density flotation in sodium-polytungstate solution (1.6 g cm$^{-3}$). After removal of the unprotected free light fraction in a first flotation step, samples were sonicated (site-specific energy application) to disrupt aggregates and separate the occluded light fraction from the targeted MOC in the HF (for details see Schrumpf et al. (2013)). Concentrations of OC in MOC are shown in Figure 1 and ranged in the uppermost layer from 16.8 g kg$^{-1}$ at Gebesee to 108 g kg$^{-1}$ at Laqueuille and from 4.6 g kg$^{-1}$ at Hesse to 44 g kg$^{-1}$ at



Laqueuille in the deepest studied layer. Selected bulk soil properties of respective samples were adopted von Schrumpf et al. (2013) and are summarized in Table 1.

We randomly selected three out of the original ten replicated soil cores per site and analyzed the soil layers 0-5 cm, 10-20 cm, and 30-40 cm (for Wetzstein it was 0-10 cm, 10-30 cm and 30-50 and soil pits instead of cores were analyzed, see Schrumpf et al. 2013) for (1) extractable MOC and (2) oxidation-resistant MOC.

Extractable MOC was determined by weighing 25 g of HF material of each sample into 250-ml polypropylene centrifuge bottles and adding 125 ml of a 1:1 solution containing 0.8 M NaF and 0.2 M NaOH. Containers were then closed and agitated overnight (at least for 18 hours) in an end-over shaker. A few drops of magnesium chloride were added as flocculant to the solution, which was then centrifuged for 15 minutes at $4000 \times g$. Then supernatants were decanted into 1000-ml PE bottles and stored in the refrigerator. Another 125 ml of the extraction solution was added to settled soil material in the

centrifuge tube, stirred and mixed well to repeat the extraction for a total of four times. Finally, the combined extract solution from each sample was passed through 90-mm glass fiber filters and stored in a 4° C climate controlled room until transfer into deionized water-rinsed, 75 cm long SERVAPOR 29-mm cellulose-acetate tubings for dialysis. The ~2/3 full tubes were placed into clean 10-l buckets filled with deionized water, which was frequently replaced until the electrical conductivity of the external solution was <2 µS. The content of the dialysis tubes was then freeze dried, and analyzed for

total C, N, and $^{14}$C signatures (as described below). The extraction residual soil containing the non-extractable OC was washed three times with deionized water to minimize remaining fluoride content before freeze drying and analyses of total C and N contents, and $^{14}$C signature (as described below).

Oxidation-resistant MOC was obtained by following a slightly modified procedure from Jagadamma et al. (2010). After letting 2 g of soil soak in 20 ml of Millipore DI water for ten minutes, 60 ml of 10% hydrogen peroxide was gradually added

to the soil. After the frothing had subsided from the reaction of wet samples with 60 ml of $H_2O_2$ at room temperature, the samples were heated and stirred regularly in a 50°C water bath in order to catalyze the oxidation of organic matter. Because $H_2O_2$ decomposes with exposure to light and temperature, the samples were centrifuged, the supernatant decanted, and fresh $H_2O_2$ added to continue the oxidation. Each sample was oxidized for two periods of 24 hours and one period of 72 hours. After the final oxidation, samples were centrifuged at $3500 \times g$ for at least 15 minutes, and then washed three times with 80

ml of deionized water. Magnesium chloride solution was added to enhance flocculation during centrifugation. After the final oxidation each sample was washed three times with DI water and afterwards freeze dried. The samples were then homogenized using a ceramic ball mill and measured for total carbon and nitrogen content by dry combustion with the Vario EL CN analyzer (Elementar Analysensysteme GmbH, Hanau, Germany). All soils analyzed in this study were free of carbonate. Therefore, total carbon measurements are equivalent to total organic carbon in the soil.

Radiocarbon contents of the samples were measured on graphitized samples at the $^{14}$C laboratory in Jena, Germany (Steinhof et al., 2017). Soil samples were weighed into tin capsules and combusted in an elemental analyzer. Samples containing carbonate were decalcified using 2 mol $l^{-1}$ HCl solution prior to combustion. The evolved $CO_2$ was transferred into a glass tube cooled by liquid nitrogen, and reduced to graphite at 600 °C under hydrogen gas atmosphere, using iron as catalyst. The



graphite was analyzed by $^{14}$C AMS (3MV Tandetron 4130 AMS $^{14}$C system: High Voltage Engineering Europe, HVEE, The Netherlands). Samples with low OC concentrations from the $H_2O_2$ residues were combusted with CuO wire in quartz tubes and graphitized using a sealed zinc reduction method, then analyzed at the WM Keck Carbon Cycle AMS facility at UC Irvine (Xu et al., 2007). Radiocarbon data are reported as $\Delta^{14}$C in per mille [‰], which is the relative difference in activity with respect to a standard (oxalic acid standard NBS SRM 4990C), after normalization to $\delta^{13}$C (fractionation correction) and correction for decay between 1950 and time of analysis (2015). The average measurement precision of the $\Delta^{14}$C values was

2.8‰.

To determine the amount of carbon in the residues from both extraction procedures, we multiplied the measured OC content in the recovered residues with its mass. The amount of carbon lost by the treatments was determined as difference between the original OC content of the HF sample and OC in the residues. The radiocarbon contents of the $H_2O_2$ residues were directly measured, and the $^{14}$C fraction of the OC lost/extracted ($^{14}C_{extract}$) was determined by mass balance as follows:

$^{14}C_{HF} = (OC_{extract}/OC_{HF}) \cdot {}^{14}C_{extract} + (OC_{residue}/OC_{HF}) \cdot {}^{14}C_{residue}$

$^{14}C_{extract} = ({}^{14}C_{HF} - (OC_{residue}/OC_{HF}) \cdot {}^{14}C_{residue})/ (OC_{extract}/OC_{HF})$

The same formula was used to determine the $^{14}$C fraction of OC extracted by NaF/NaOH. Since OC extracted by NaF/NaOH extraction was also measured directly, the mass balance results allow for identifying potential bias in measured $^{14}$C data caused by losses of extracted OC during dialyses.

Solid-state cross-polarization magic angle spinning $^{13}$C-nuclear magnetic resonance (CPMAS $^{13}$C-NMR) spectra of NaF/NaOH extracted, dialyzed, and freeze-dried OM were recorded on an Avance III spectrometer (Bruker BioSpin GmbH, Rheinstetten, Germany) at a resonance frequency of 100.5 MHz, with a proton spin-lock and decoupling frequency of 400 MHz. The proton nutation frequency was 80 kHz, corresponding to a π/2 pulse duration of 3.12 μs. The cross-polarization time was 500 μs. Samples were weighed into 4-mm zirconium oxide rotors that were spun at 10 kHz around an axis declined

by the 'magic angle' of 54.74° against the static magnetic field; contact time was 1 ms and the recycle delay time was set to 0.4 s. Depending on the sample, between 4,000 and 35,000 scans were recorded. The spectra were processed with a line broadening of 100 Hz. Chemical shifts are given relative to the resonance of tetramethylsilane. After baseline correction, the intensities of spectral regions were corrected for different cross-polarization efficiencies in different spectral regions. To do so, $^1H$ $T_{1\rho}$ as well as $T_{CH}$ were estimated selectively for the regions. Since OM samples with C contents >30% were analysed,

no treatment for removal of paramagnetic mineral phases was required and the obtained spectra were well resolved and showed no indications of paramagnetic interferences. In addition, we analyzed one bulk MOC sample low in pedogenic oxides without removal of paramagnetic phases. Also that spectra was reasonably well resolved and without indications of paramagnetic interferences.

Resonance areas were calculated by electronic integration: alkyl region (0–50 ppm), mainly representing C atoms bonded to

other C atoms (methyl, methylene, methine groups); O/N-alkyl region (50–110 ppm), mainly representing C bonded to O and N (carbohydrates, alcohols, and ethers) and including the methoxyl C (peak centred around 56 ppm); aromatic region (110–160 ppm), representing C in aromatic systems and olefins, and (d) the carbonyl region (160–220 ppm), including





carboxyl C (160–190 ppm). Further information on the assignment of [13]C-NMR regions are given by Wilson (1987) and Orem and Hatcher (1987).

## 3 Results

### 3.1 Mineral-associated organic carbon removed by NaF/NaOH and $H_2O_2$

The NaF/NaOH extraction removed on average 58±11% of bulk MOC across sites and soil depths (Figure 2). With average values of 57±7% in the uppermost, 60±15% in the intermediate, and 56±11% in the deepest analyzed soil layer, there was no trend in the extraction efficiency with soil depth. Extraction efficiency was, however, on average somewhat smaller at site Gebesee than at the others sites (41±10% vs. 62±11% on average across depths). As they represent similar portions of bulk MOC, the amounts of extracted and residual OC were strongly correlated to bulk MOC concentrations across sites and soil depths (r=0.96, p<0.01; Figure 3, Figure A1). Neither contents of pedogenic oxides, OC loadings of minerals, nor soil pH affected the portion of MOC that was extractable.

On average, only 11±6% of MOC resisted to treatment with heated $H_2O_2$, with 90±8% being removed in the uppermost, 90±6% in the intermediate, and 87±5% in deepest studied soil layer (Figure 2). Accordingly, an insignificantly smaller portion of total OC was extracted in the subsoil than in the topsoil. The portion of MOC resisting the $H_2O_2$ treatment differed between study sites. The largest portion was found at the Wetzstein site (on average 19±4%), especially in the deepest layer, while smallest portions of $H_2O_2$-resistant MOC occurred at Gebesee (5±3%). The absolute amount of $H_2O_2$-resistant OC was correlated with the bulk MOC concentration (r=0.76, p<0.01; Figure 3), however, the correlation was weaker than those found residual OC after NaF/NaOH extraction.

### 3.2 NMR spectroscopy

The NMR spectra of extracted OM from soils of the sites Hainich, Hesse, and Laqueille were remarkably similar and dominated by signals in the O/N-alkyl C region (62-74% of total peak area across sites and depths) and the alkyl C region (18-25%), suggesting a strong contribution of carbohydrates and aliphatic compounds to the extracted OM (Figure 4, Table 2). All six spectra also reveal a distinct peak centered around 174 ppm, due to carboxyl C, which is in line with the fact that the alkaline extraction tends to release preferentially acidic compounds (5-9%). All spectra show small signals in the aromatic regions centered around 150 ppm (phenols) and 130 ppm (non-substituted aromatic systems). All six spectra featured signals, some even well-resolved, around 56 ppm, indicating the presence of methoxyl C.

The spectra obtained on OM extracted from the Chernozem-type soil at site Gebesee resembles those of the sites Hainich, Hesse, and Laqueille, except for that they indicate more non-substituted aromatic systems, which is in accordance with findings on the occurrence of pyrogenic OM in such soil. The spectra obtained on extracted OM from the Podzol-type soil at Wetzstein showed the strongest deviation in spectral features. The signals due to carbonyl/carboxyl, aromatic and alkyl C are much more prominent than in all other spectra.



When comparing the spectra of OM from top- and subsoils, differences were surprisingly small at Hainich, Hesse, Laqueille,
and Gebesee, with tendencies of decreasing proportions of alkyl and aromatic C in favour of increased signals of O/C-alkyl
C. The contribution of signals of carbonyl C remained fairly constant with soil depth. The change in composition with depth
was much more evident for the site Wetzstein. Here, aromatic and especially alkyl C decreased while O/C-alkyl increased,
probably reflecting the strong re-distribution of OM along the profile during podzolization. The composition of the subsoil
OM at Wetzstein approached that of the OM at the sites Hainich, Hesse, and Laqueille.

The bulk MOM and the NaF/NaOH extraction residue of the Hanich 0-5 cm sample were additionally analyzed. The signal-
to-noise ratio of these spectra was less than for the extracted OM due to the presence of paramagnetic minerals (Figure 5).
Nevertheless, the results reveal larger portions of carbonyl/carboxyl C and especially aromatic C but less O/N-alkyl C for the
extraction residue than for the extracted OM (Figure 5, Table 3).

### 3.3 Radiocarbon content of MOC fractions

The radiocarbon contents were always larger in the extracted or oxidized fractions than in the residues, and decreased
(except for the $H_2O_2$ residues at Gebesee) at all sites with soil depths (Figure 6). Decrease in [14]C contents with soil depth was
strongest for Hainich and Laqueuille, and least for Wetzstein and Gebesee. Figure 7 shows that results for directly measured
[14]C in dialyzed NaF/NaOH extracts were overall comparable to calculated [14]C in extracts from the mass balance approach,
suggesting that there were no systematic losses of older or younger C during the extraction and subsequent dialysis
procedure. For Gebesee, the mass balance approach suggests that some young carbon was probably lost during dialyses of
the extracts.

The [14]C contents of the OC extracted from the uppermost layers increased in the order Gebesee < Wetzstein (0-10 cm) <
Laqueuille < Hainich < Hesse from -126‰ to 142‰. The average difference in [14]C between extracted and residue OC was
79±36‰ across sites and increased in the order Gebesee (34±4‰) < Laqueuille (38±6‰) < Hainich (63±3‰) < Hesse
(84±5‰) < Wetzstein (100±15‰). As indicated by the almost parallel shifts in [14]C contents (Figure 6), there was no general
trend for increasing or declining differences in [14]C contents with soil depths. Instead, [14]C contents in extracts and extraction
residues were highly correlated ($r^2$=0.91, p<0.01, supplementary Figure S2). [14]C contents of bulk MOC, extractable or
residual MOC were, however, all unrelated to total MOC or its extractability (results not shown).

Residues of the $H_2O_2$ treatment were much older than residues of the NaF/NaOH extractions (Figure 6). The average [14]C
contents of $H_2O_2$ residues ranged between -36±8‰ (Laqueuille) and -691±21‰ (Gebesee) in the uppermost, and between -
310±73‰ (Wetzstein) and -630±70‰ (Hainich) in the deepest layer (Figure 6). The difference between [14]C contents of
oxidized and residual OC in the uppermost layer increased in the order Laqueuille = Wetzstein (115‰) < Hesse (239‰) <
Hainich (290‰) < Gebesee (591‰), and increased slightly with soil depth at Hesse, Hainich, and Wetzstein.

When comparing [14]C contents of OC removed by either NaF/NaOH or $H_2O_2$ treatments, we found, surprisingly, that there
was basically no [14]C-difference for most sites (Figure 8), despite different total amounts of total OC removed by the
individual procedures. As indicated by the almost parallel shift in [14]C contents of $H_2O_2$ residues from NaF/NaOH residues in





all soil profiles, both were typically highly correlated within profiles (Figure 8). While [14]C in NaF/NaOH residues of the sites Wetzstein and Laqueuille deviated from $H_2O_2$ residues by only 62±26‰, it was 258±99‰ for Hesse and Hainich (Figure 8). The only exception was again Gebesee, where OC extracted by NaF/NaOH was on average younger than OC removed by

$H_2O_2$, and [14]C contents of the two residues were not correlated and differed on average by 456±135‰.

## 4 Discussion

### 4.1 Unexpected similarity of the NaF/NaOH-extractable portion of total MOC

Strong hysteresis, rendering part of adsorbed OC resistant to desorption, is a common phenomenon found in sorption-desorption experiments with OM (Gu et al., 1994; Oren and Chefetz, 2012). Given that desorption rates into ambient soil

solutions are small, we applied NaF/NaOH extraction, operating via the combination of competing $OH^-$ and $F^-$ anions, and alkaline conditions, as indicator for potential maximal desorption. Accordingly, the method targets OC bound to minerals by Coulombic forces and surface complexation, but likely includes OC held by different more weakly forces, such as hydrogen bonds, cation bridges, or hydrophobic interactions.

We hypothesized that the portion of extractable C would increase with OC loading of minerals and accordingly be higher in

topsoils with larger MOC concentrations than in subsoil layers with smaller ones. Since minerals have different characteristic dominant binding modes for OC at a given pH (Mikutta et al., 2007), we further assumed that extractability would depend on mineral composition and soil pH, with smallest desorption in acidic soils with large contents of pedogenic oxides. And finally, we expected that land use- and site-specific differences in OM quality would influence MOC extraction. Our results showed, however, that for most test sites a surprisingly constant portion of on average 62±11% of the MOC was

extracted by NaF/NaOH, irrespective of soil depth, study site, and original OC concentration. Only at the site under arable management, Gebesee, the portion of extractable OC was smaller (41%). This could be due to the depletion in weakly bound MOC in response to reduced input and accelerated mineralization of OC caused by the constant soil mixing typical for agricultural sites (Plante et al., 2005; Helfrich et al., 2007). For less disturbed soils with natural vegetation our results suggest that despite the presumed variation in the chemical composition of OC and the mineral assemblage, actual interactions

between them seem rather uniform.

Experiments in the laboratory showed similar extractability of OC in NaF/NaOH from two Fe-oxides, despite differences in the absolute amounts of OC sorbed by goethite and ferrihydrite (Kaiser et al., 2007). The extractability from these model MOC was also unexpectedly similar to results of this study (around 65%, Kaiser et al., 2007), and OC extracted with NaOH from experimentally produced MOC on goethite in another experiment (57-60%, Kaiser and Guggenberger, 2007).

Nevertheless, we expected to see a greater MOC extractability at sites poor in oxides but rich in clay minerals, with a hypothesized larger share of more weakly bound OC on MOC. It is possible that some of the weakly bonded OC was already lost during the preceding density fractionation with Na polytungstate solution (Schrumpf et al., 2013). The missing relation between soil mineralogy and extractability suggests that predominantly OC bound by the same mechanism was extracted,



irrespective of mineral composition. This indicates that either the same dominant bond mechanism operates for different

minerals, or that extracted OC originated predominantly from one mineral type. Pedogenic oxides as well as clay minerals can hold adsorbed OC by covalent bonds (e.g. Chen et al., 2017; Gu et al., 1994). These bonds are formed between metal-coordinated hydroxyl groups exposed at surfaces of pedogenic oxides and clay mineral edges and carboxyl groups of OM. The lacking variation in extractability of MOC can thus reflect the dominant role of this type of sorptive interaction in the formation of mineral-organic associations under the acidic to neutral soil reactions of the study sites. The linear relation

between MOC and the sum of oxalate-extractable Al and dithionite-extractable Fe in our study indicates that pedogenic Al and Fe oxi-hydroxides were important for OC binding across the study sites (Figure A2). Apparently, even very high clay contents (>50%), as at the Hainich site, cannot compensate for smaller contents of pedogenic oxides for MOC storage. The measured uniform extractability of MOC could accordingly also be an immediate result of the pedogenic oxides controlling MOC accumulation.

Despite its different chemical composition, desorption of MOC from the coniferous forest site Wetzstein did not differ from desorption of MOC from the other non-arable sites. This indicates that the overall molecular composition of OM matters less for sorption-desorption than the presence of functional groups capable to interact and form bonds with mineral surfaces. This is in agreement with the idea discussed above, that extracted MOC was predominantly from the same bond type, and rather driven by the presence of functional groups on both, minerals and OM, than by their type or composition.

The very similar composition of MOC extracted from the sites Hainich, Hesse, and Laqueuille hints also at uniforming processes within the mineral soil, again with no variations with differing contents of pedogenic oxides. One explanation could be that the type of sorptive interaction targeted by the extraction used selects for a specific composition of the extracted MOC. Alternatively, the similar MOC composition may reflect the uniforming microbial processing of the organic input (Liang et al., 2017). However, differing composition at the sites Gebesee and Wetzstein suggest that specific site and

pedogenic properties can nevertheless be imprinted in MOC composition. For the Chernozem at Gebesee, higher contents of non-substituted aromatic systems are probably due to a different vegetation history, where also fire played a role. Differences in OM composition between the sites Wetzstein and Laqueille were somewhat surprising, since the pedo-environmental conditions (acidity and mineralogy) of OM accumulation in Podzol-type and Andosol-type soil are often considered similar (Aran et al., 2001; Young et al., 1980). The decomposing conditions at the Podzol-type soil (pH,

microbial community, conifer-dominated vegetation, soil climate) probably affected the MOC composition.

    Experiments on model minerals suggested that desorption increases with increasing OC loading of minerals (Kaiser and Guggenberger, 2007). This would be in line with our observation of reduced desorption at the agricultural soil Gebesee, where OC loading of minerals is probably reduced relative to undisturbed soils by smaller inputs, increased mineralization, and soil mixing by plowing. Although the OC loading of minerals is smaller in subsoils, soil depth did not affect MOC

extractability in this study. Also Mikutta et al. (2010) and Möller et al. (2000) observed no significant increase in NaF/NaOH extractability of OC with soil depth. One reason could be that DOC input to subsoils probably occurs mostly along specific flow paths (Bundt et al., 2001) so that exposed mineral surfaces in subsoils could be similarly loaded with OC as topsoils,





resulting in similar desorption. Additionally, increasing pH with soil depth possibly reduces the sorption capacity of minerals in subsoils. In any case, an overall consequence of our findings is that a decline in potential desorption is not responsible for greater subsoil OC stability (Rumpel and Kögel-Knabner, 2011) and cannot explain the typically observed increase in carbon ages of MOC with soil depths.

One question remaining is what characterizes the non-desorbable MOC fraction. While most aggregates were destroyed by the sonication treatment during density fractionation, potential contribution of stable microaggregates to OC protection against extraction cannot be excluded. Since desorption of OC with NaOH did not result in an increase in the micropore volume, and because non-desorbable OC had a higher apparent density, Kaiser and Guggenberger (2007) ascribed the MOC fraction not extractable into NaOH to OC tightly bound to mineral surfaces by multiple bonds, preferably at the edges of and across micropores. Since multiple bonds require a higher number of functional groups involved, OC of extraction residues should be enriched in carboxylic groups. This is supported by our NMR data, showing a higher share of carbonyl/carboxyl groups in non-extractable than in extractable MOC from the Hainich site. Phenolic and aromatic groups, which are also known to bind preferentially and strong to mineral surfaces (Chorover and Amistadi, 2001; Kothawala et al., 2012; Avneri-Katz et al., 2017), were also enriched in the residues, while O/N-alkyl-C was depleted. The non-desorbable portion of OC could accordingly be composed of aromatic and other compounds strongly bound to mineral surfaces by multiple functional groups.

## 4.2 Missing relations between OC desorption and [14]C-age

Results of the [14]C analyses of NaF/NaOH extracts and residues confirmed our hypothesis of preferential extraction of younger carbon. That result was consistent across soil types and depths. It suggests that desorption facilitates the exchange of old for new OC on mineral surfaces, resulting in on average younger extractable than non-extractable OC, which is less frequently exchanged. Using Na pyrophosphate as extractant, Heckman et al. (2018) also observed consistently younger OC in MOC extracts and older OC in extraction residues across different soils and soil depths. Na pyrophosphate has a similar effect on MOC as NaF/NaOH due to a comparable raise in pH and because both, pyrophosphate and fluoride, act as chelating agents, and thus, strongly compete with OC for binding sites on mineral surfaces.

We further hypothesized that soils with more extractable MOC should have on average younger [14]C ages. Such a positive relation between extractable OC and [14]C was not observed for any soil depth or fraction across sites. This is possibly a direct consequence of the rather small variation in OC extractability between samples. Further, some laboratory studies showed that OC sorbed to goethite is less bioavailable than OC on clay minerals (e.g. Mikutta and Kaiser, 2011; Mikutta et al., 2007). This should lead to faster turnover of MOC in clay-rich sites relative to sites rich in oxides, and so we expected to find younger extractable MOC at clay-rich than at oxide-rich sites. By contrast, we also found no indication that clay-rich soils have generally younger carbon than soils rich in pedogenic oxides. Again, the assumptive dominant control of pedogenic oxides on total MOC contents across sites could have masked a potential effect of mineral composition on bioavailable MOC.





Since [14]C contents of bulk MOC and its fractions are apparently independent of OC resistance to desorption and mineralogy, site specific differences are possibly rather driven by OC input. Assuming that DOC is either the main direct source for new MOC or the carbon source of microbial residues sorbing to minerals, its concentration, composition, and flux in the soil solution could determine OC accumulation and exchange rates on minerals (Kaiser and Kalbitz, 2012; Sanderman et al.,

2008). Similar to the chemical composition of the extractable MOC, also its age could be controlled by DOC production, and thus, the overall decomposition conditions and ecosystem properties. The idea is supported by the positive relation between the [14]C contents of NaF/NaOH extracts and [14]C contents of the light fractions in topsoils (Figure 9). Site-specific differences in MO[14]C are, therefore, possibly dependent on local microbial activity and litter decay rates. MOC ages can further be modified by site- and soil depth-dependent differences in the [14]C content of DOC. The thick humus layer (8-14 cm)

overlying the topsoil of the Podzol at Wetzstein could for example have contributed to the old extractable MOC in the uppermost mineral layer via input of pre-aged DOC.

**4.3 Comparison between NaF/NaOH extraction and $H_2O_2$ oxidation results**

In accordance with our expectation, residues of $H_2O_2$ treatments were older than residues of NaF/NaOH extractions. We further hypothesized that, as a result of the removal of a larger, and therefore on average more stabilized fraction with the

heated $H_2O_2$ treatment, OC oxidized would also be on average older than extracted OC. Despite $H_2O_2$ removed on average 89% of bulk HF-OC and NaF/NaOH removed only 62%, both removed MOC fractions had comparable [14]C ages. Seemingly, NaF/NaOH residues still contain oxidizable OC of similar or only slightly older age as the extracted material. This result indicates that both, NaF/NaOH and $H_2O_2$, removed mostly OC from a younger, [14]C-richer pool, leaving increasingly old residual OC behind, the more OC is removed. While unexpected, Jagadamma et al. (2010) also observed similar [14]C contents

of OC removed from bulk soil samples with different oxidation reactants, irrespective of the extent of OC removal. Applying a mass balance approach to the results of the different extraction procedures applied by Helfrich et al. (2007) shows that also for their soils [14]C contents of removed OC were similar, irrespective of the extracted OC amounts. Accordingly, a large portion of MOC could be more homogenous in [14]C contents than expected, while apparently only a rather small portion has very old ages.

The [14]C difference between NaF/NaOH and oxidation residues was unexpectedly smaller for the two soils rich in pedogenic oxides (Laqueuille, Wetzstein) than for the other sites. The soils rich in pedogenic oxides also had slightly higher amounts of OC left in oxidation residues, suggesting that they protected a larger portion of OC against oxidation, but this OC was younger than in the soils from the other sites. Eusterhues et al. (2005) similarly observed that more OC resisted $H_2O_2$ oxidation in subsoils rich in pedogenic oxides, and older residues in the Dystristic Cambisol than the Haplic Podzol studied.

Accordingly, high contents of pedogenic oxides in soils seem to increase oxidation resistance of MOC but do generally not increase residue ages. The comparatively young oxidation residues at Wetzstein and Laqueuille in our study could be due to high DOC fluxes, and thus, overall faster OC replacement of all MOC components at Wetzstein, and the younger soil age of the Andosol soil at Laqueuille. It is further possible that at some of the sites, old oxidation-resistant OC was inherited from



the parent material (e.g., the loess layer or limestone residues in Hainich and Hesse) or a specific fire history (e.g., at the
Chernozem site Gebesee, where only small, but very old amounts of OC were left after the $H_2O_2$ treatment). The observed
increase in $^{14}C$ differences between oxidized and residual OC with smaller OC amounts left in $H_2O_2$ residues across sites
(Figure A3) suggests that there could also be a site- and depth-independent trend for increasingly older OC the smaller the
OC amount left on the minerals.

**4.4 Changes in OC turnover along soil profiles**

The close correlations between $^{14}C$ contents of NaF/NaOH extracts and residues along the soil profiles (see also Figure A4)
result in a parallel decline in $^{14}C$ of both fractions with soil depths. Accordingly, not only the same portion of OC was
extractable across soil depths, but also absolute differences between $^{14}C$ contents of extracts and residues remained constant.
In line with the observed constant extractability, $^{14}C$ depth profiles of MOC are therefore apparently not driven by a specific
increase in the stability of either residual or extractable OC. Instead, the same extraction-sensitive bond type and strength
was affected along soil profiles, independent of site-specific differences in absolute $^{14}C$ values or the slope of the depth
decline in $^{14}C$. This suggests that (1) the distribution of MOC between fast and slower cycling OC is constant with depth and
that (2) the overall shape of the $^{14}C$ distribution within a sample (if we consider $MO^{14}C$ to be a continuum) remains constant.
This is supported by the observation that also residues of the $H_2O_2$ treatment, though on average much older, declined almost
parallel to NaF/NaOH residues with soil depth. Whatever causes the $^{14}C$-decline with soil depth, it is apparently shifting the
entire $^{14}C$ age distribution of the MOC. This hints, similar to the uniform extractability, at subsoil MOC being similarly at
equilibrium with its environment than topsoil MOC.

Differences in the overall depth decline of $^{14}C$ between sites could be due to differences in (D)OC transport rates along the
soil profile or in root litter input. Kindler et al. (2011) measured overall much higher DOC leaching rates from topsoils at
Wetzstein than at Laqueuille or Hainich, while leaching rates from Wetzstein subsoils were only slightly increased,
suggesting that a large portion of this mobilized topsoil DOC at Wetzstein was adsorbed in the subsoil. This probably
resulted in a rejuvenation of subsoil OC, and thus, the less steep depth decline of $^{14}C$ relative to the other two sites.

The small increase in the portion of oxidation-resistant OC at depth together with the slightly stronger decline in $^{14}C$ of
oxidation residues with depth relative to bulk MOC, indicate that, different from desorption resistance, oxidation resistance,
i.e., chemical recalcitrance, of a small portion of MOC becomes a bit more relevant to the formation of stable MOC with soil
depth.

**5 Synthesis and implications**

The initial assumption of this study was that the stability of mineral-bound OC should be related to desorption, and thus,
vary between soils with different mineral composition and under different land use. It turned out that OC extracted in
NaF/NaOH was indeed consistently younger than bulk MOC, suggesting that desorbed OC was more frequently exchanged





than the older residue. The extractability of MOC was, however, uniform across non-cultivated soils and depths for the acid to neutral central European soils studied, irrespective of mineral composition or chemical composition of extracted OC. Total MOC amounts were controlled by contents of pedogenic Al and Fe oxides, irrespective of the clay content in the samples. Therefore, extraction results probably reflect only the response of oxides, which then conceal potential mineral-specific differences in binding strengths observed for pure minerals in the laboratory. This is supporting the paradigm that

oxides are more important than clay for OC storage in soil (Rasmussen et al., 2018) and could facilitate easier modelling of MOC formation and turnover in the future.

The overall chemical uniformity of extracted MOC across sites and depths suggests selection for specific OC molecules during MOC formation. Alternatively, microbial processing of OC on mineral surfaces homogenizes MOC composition relative to original OM differences from litter and vegetation types. Small differences in the amounts of e.g. aromatic

compounds in extracted MOC with soil depth and between sites still indicate the contribution of strongly sorbing compounds to MOC formation. However, the formation of extractable MOC seems to be overall controlled by the presence of interactive functional groups, such as metal-coordinated hydroxyl groups on the minerals and carboxyl groups of OM, while the minerals or OM molecules, these groups are attached to, are rather interchangeable.

Despite the overall similarity in MOC composition and extractability across sites, $^{14}$C contents of extracted MOC (and of

extraction residues) were proportional to bulk MOC, and thus, exhibiting site-specific differences. In contrast to total MOC storage, its $^{14}$C content seems not controlled by sorbing minerals but rather by the turnover, accumulation, and displacement rates of OC. This would imply that MO$^{14}$C should be sensitive to the $^{14}$C content and amount of DOC entering or leaving MOC, and thus, ecosystem properties driving OM decomposition, DOC production and transport (such as litter or OM amount and chemistry, pH, microbial community, climate).

No indication for the presence of stronger, less desorbable bonds between OC and minerals in subsoils than topsoils was observed, and there was no preferential decline in the $^{14}$C content of extracts or extraction residues with depth. Consequently, most subsoil OC would also not be better protected against desorption, and thus, potential subsequent degradation than topsoil OC. Reduced desorption observed at the cropland site indicates that it might not be totally unrelated to OC-loading of minerals. A possible explanation for the missing depth gradient in MOC extraction is that subsoil MOC is typically similarly

at equilibrium with the local conditions as the topsoil. This could be the case if OC input to undisturbed subsoils occurs mostly in hotspots via root input and along preferential flow paths, where OC is binding to exposed mineral surfaces but bypassing surfaces located inside aggregates. Since difference in desorption cannot explain different MO$^{14}$C depth distributions, these are then probably rather due to variations in OC input by roots and DOC transport. Under conditions of much lower direct litter input, vertical OC transport according to the "cycling downwards" concept (Kaiser and Kalbitz,

2012) would become more important for $^{14}$C in subsoils, thereby shaping the different depth profiles.

NaF/NaOH extraction and H$_2$O$_2$ oxidation both suggest that the largest part of total MOC has similar $^{14}$C contents, irrespective of the way it is removed, while only a small portion (<20%) of total MOC is much older. Different chemical fractionation schemes apparently always remove OC from the same continuum, leaving increasingly old OC behind. Future



research on the role of DOC and soil solution chemistry for accumulation and exchange rates of OC on mineral surfaces

along soil profiles under field conditions might help to better understand the emergence of the age continuum of MOC.

**Author contributions**

MS, KK, AM and ST designed the experiments and AM carried them out. GH conducted the NMR analyses. MS prepared

the manuscript with contributions from all co-authors.


**Acknowledgements**

We are grateful to the Routine Measurements and Analyses group (Roma, Ines Hilke and Birgit Fröhlich), and the [14]C

Analytik group (Axel Steinhof, Heike Machts) of the Max-Planck Institute for Biogeochemistry and Xiaomei Xu (UC

Irvine) for their help with sample analyses.

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

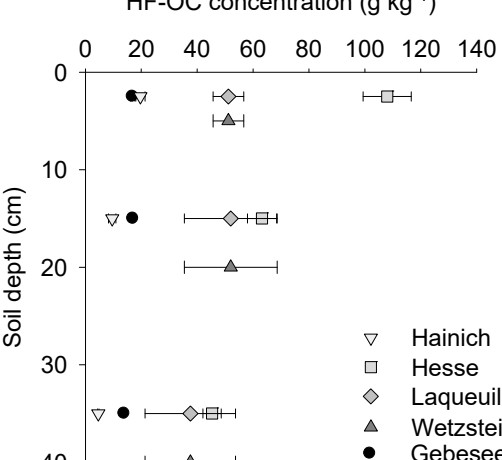

**Figure 1: Figure 1: Original concentration of OC in mineral association of the studied soil samples from 5 sites (data adopted from Schrumpf et al. 2013).**





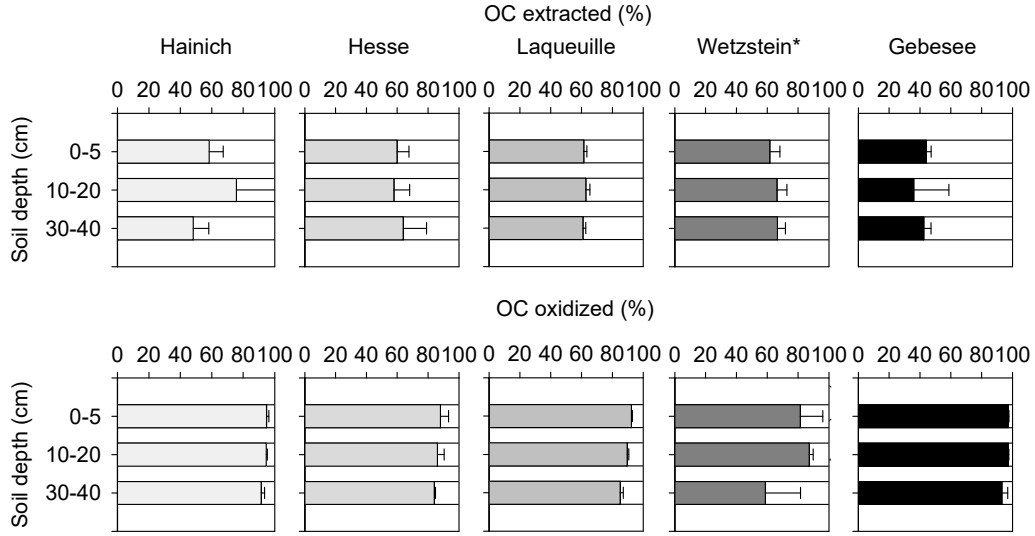


**Figure 2: Portion of OC extractable with NaF/NaOH or oxidized with heated $H_2O_2$ from the mineral associated OC fractions of soil samples from five different sites (Hainich, Hesse, Laqueuille, Wetzstein, Gebesee) and three soil depths (0-5 cm, 10-20 cm, 30-40 cm); for Wetzstein the studied soil depths were 0-10 cm, 10-30 cm, and 30-50 cm.**

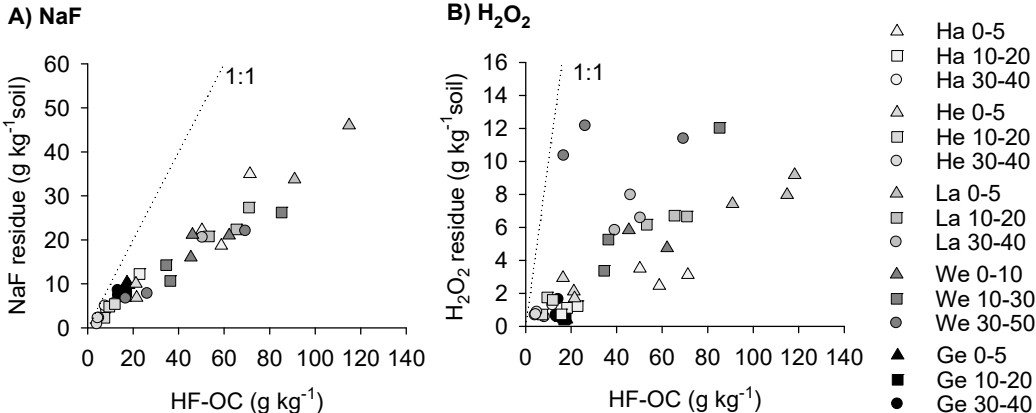

**Figure 3: Dependency of residual OC from original concentrations of mineral associated OC for A) NaF/NaOH extraction (left) or B) the $H_2O_2$ oxidation (right) for all study sites (Hainich (Ha), Hesse (He), Laqueuille (La), Wetzstein (We), Gebesee (Ge)).**
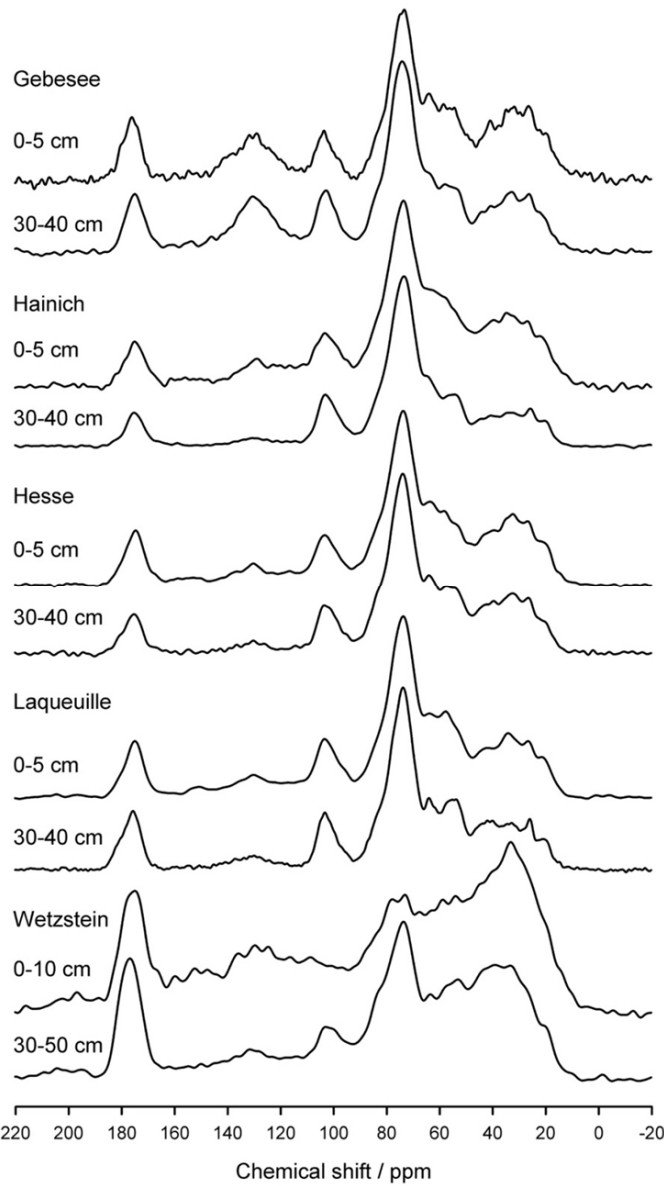

**Figure 4: NMR spectra of OM extracted into NaF/NaOH from the mineral associated fraction of two soil depths from the five study sites Gebesee, Hainich, Hesse Laqueuille, and Wetzstein.**





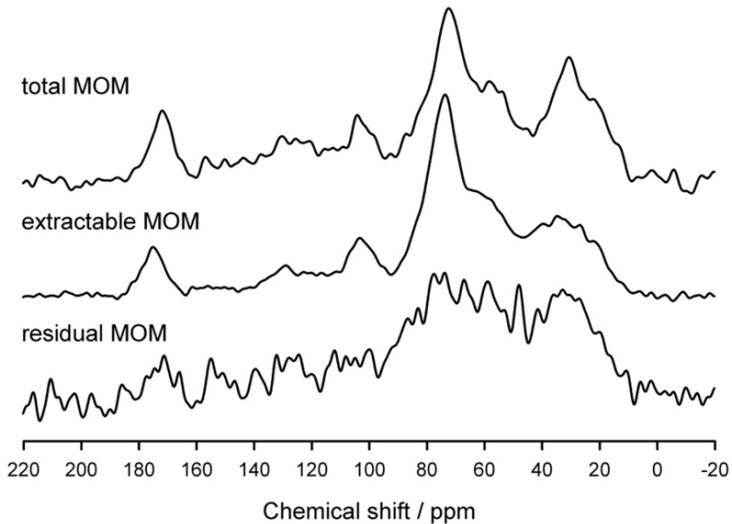


**Figure 5: NMR spectra of OM from the 0-5 cm layer of the sites Hainich. Top: total mineral associated OM (MOM), middle: MOM-fraction extracted into NaF/NaOH, bottom: residual OM after extraction of MOM into NaF/NaOH.**





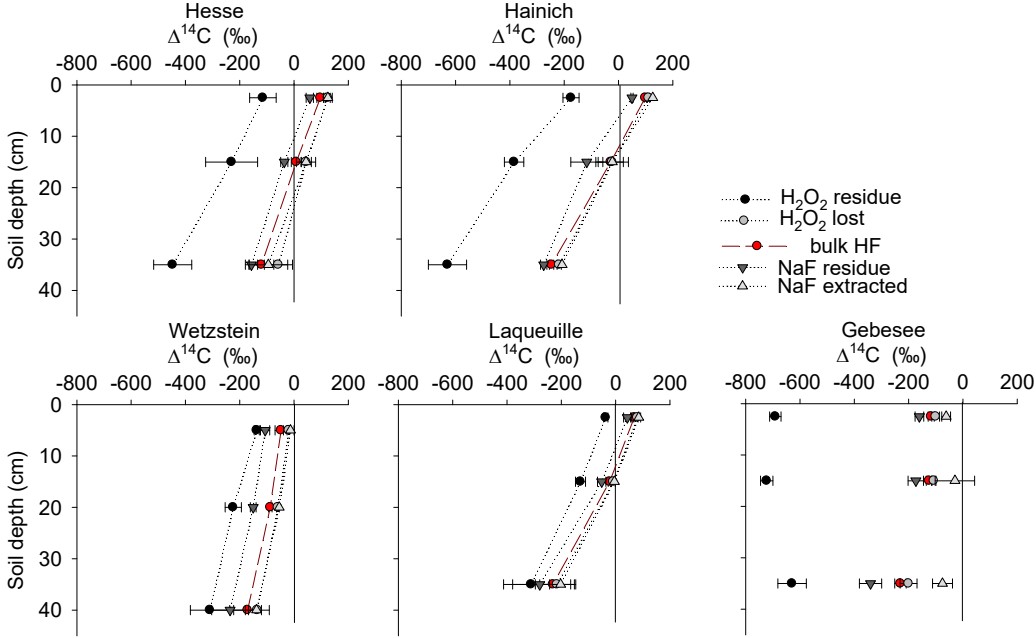


**Figure 6: Depth profiles of radiocarbon (Δ¹⁴C) in bulk mineral associated OC (bulk HF), as well as in OC removed from mineral surfaces using either NaF/NaOH or H₂O₂, and in the respective OC residues remaining on mineral surfaces.**





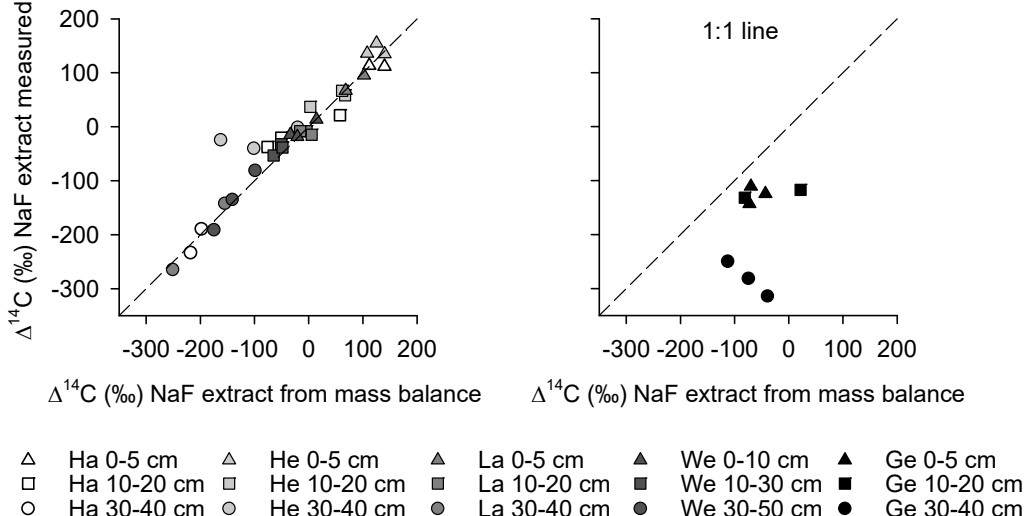

**Figure 7: Comparison of $^{14}$C contents of NaF/NaOH extracted OC obtained using a mass balance approach and from direct measurements of the extracts after dialyses (study sites: Hainich (Ha), Hesse (He), Laqueuille (La), Wetzstein (We), Gebesee (Ge)).**

**Comparison of $^{14}$C in extracts**

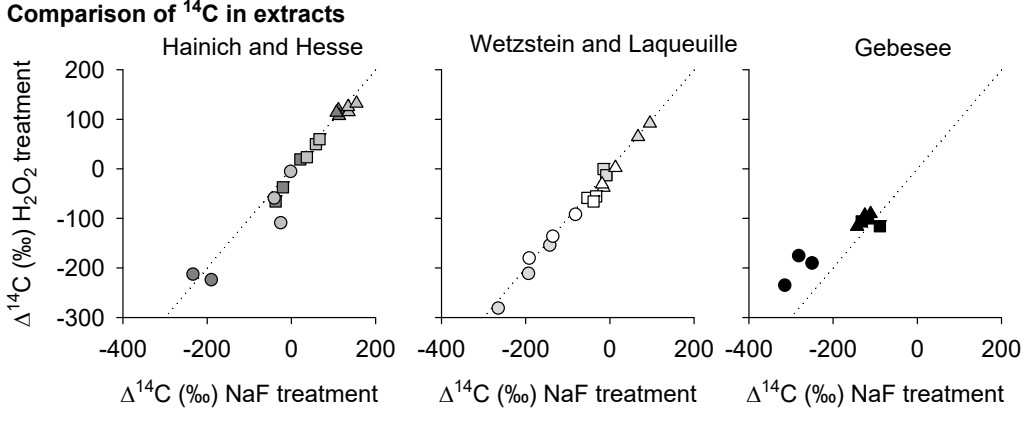

**Comparison of $^{14}$C in residues**

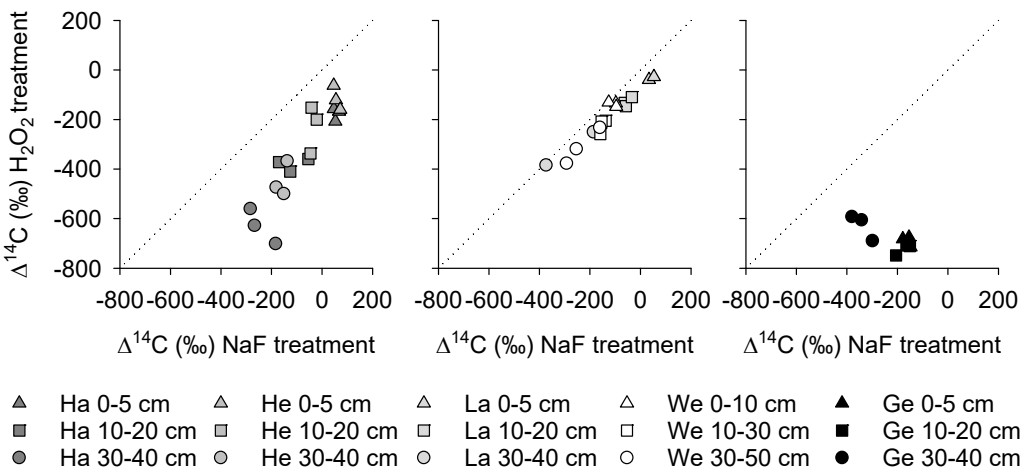

**Figure 8: Relation between the radiocarbon content of OC removed from mineral surfaces by H$_2$O$_2$ and NaF/NaOH (upper graphs) and between the radiocarbon content of OC residues after treatment with H$_2$O$_2$ and NaF/NaOH (lower graphs) for the study sites Hainich (Ha), Hesse (He), Laqueuille (La), Wetzstein (We), and Gebesee (Ge).**





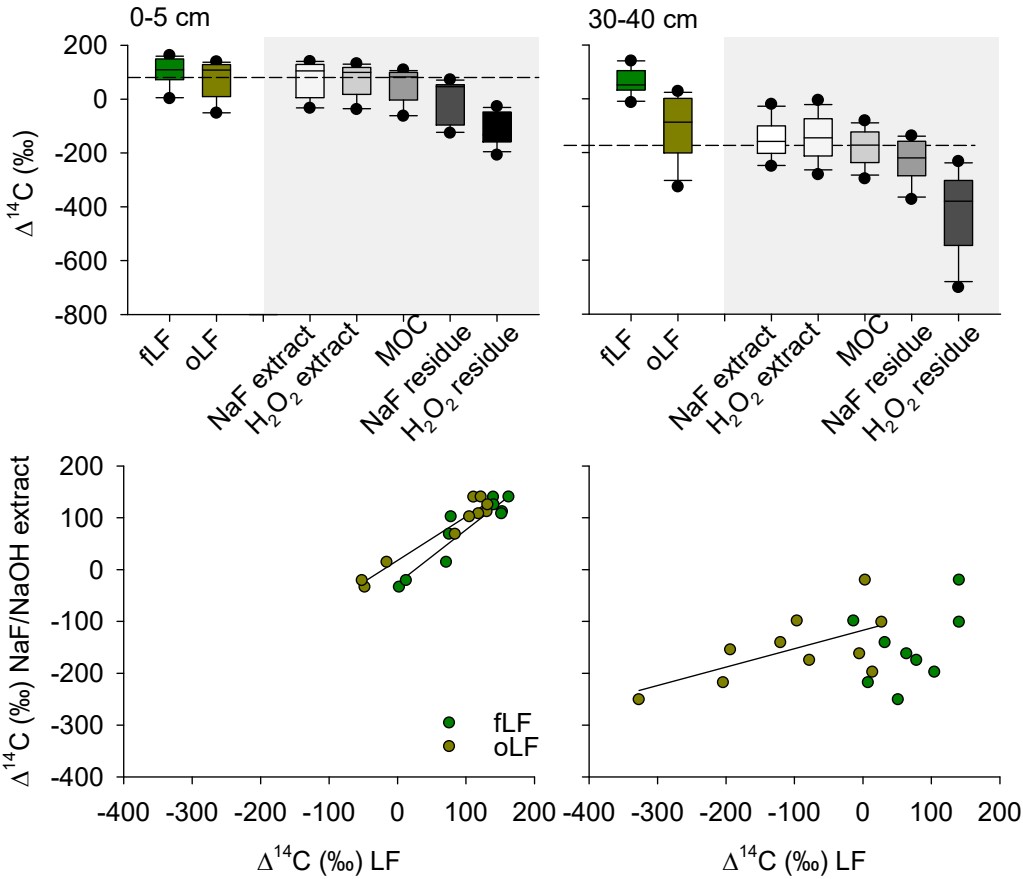

**Figure 9: Top:** box-plots giving an overview of $^{14}$C contents in all OC fractions separated from bulk samples in 0-5 cm (left) and 30-40 cm (right) depth for the non-arable samples (dashed line denotes the median of the $^{14}$C in bulk MOC). **Bottom:** relation between $^{14}$C contents of OC in NaF/NaOH extracts of MOC and OC in the two light fractions in 0-5 cm (left) and 30-40 cm (right) depths for the non-arable samples. fLF: free light fraction, oLF: occluded light fraction, values adopted from Schrumpf et al. (2013).





**Table 1: Basic soil properties of the study sites (means of the three replicated samples analyzed for mineral associated carbon per soil depth with standard deviations in brackets). Values selected cores are adopted from Schrumpf et al. 2013.**


| Site and soil depth | pH (H$_2$O) | OC g kg$^{-1}$ | CN | Ci g kg$^{-1}$ | Sand g kg$^{-1}$ | Clay g kg$^{-1}$ | Feo g kg$^{-1}$ | Fed g kg$^{-1}$ | Alo g kg$^{-1}$ |
|---|---|---|---|---|---|---|---|---|---|
| **Hainich** | | | | | | | | | |
| 0-5 | 6.1 (0.9) | 73 (11) | 13.0 (0.5) | 0 | 22 (4) | 546 (5) | 3.0 (1.0) | 14.0 (0.8) | 1.9 (1.1) |
| 10-20 | 6.7 (0.5) | 27 (6) | 10.7 (0.4) | 0 | 28 (4) | 514 (5) | 2.0 (0.4) | 15.3 (1.0) | 1.8 (0.6) |
| 30-40 | 7.4 (0.2) | 11 (1) | 9.0 (0.4) | 0 | 23 (7) | 731 (7) | n.d. | n.d. | n.d. |
| **Hesse** | | | | | | | | | |
| 0-5 | 4.6 (0.7) | 31 (6) | 13.9 (0.6) | 0 | 68 | 342 | 1.8 (0.3) | 11.8 (1.3) | 1.1 (0.2) |
| 10-20 | 4.5 (0.3) | 14 (4) | 11.8 (0.5) | 0 | 56 (12) | 315 (11) | 1.4 (0.5) | 11.5 (1.1) | 1.0 (0.1) |
| 30-40 | n.d. | 6 (0) | 8.3 (0.0) | 0 | 55(3) | 371 (22) | n.d. | n.d. | n.d. |
| **Laqueuille** | | | | | | | | | |
| 0-5 | 5.3 (0.2) | 126 (11) | 11.1 (0.2) | 0 | 186 (36) | 263 (28) | 12.7 (0.1) | 24.0 (0.6) | 19.3 (0.7) |
| 10-20 | 5.6 (0.3) | 66(7) | 10.2 (0.2) | 0 | 259 (58) | 215 (8) | 16.4 (3.0) | 20.3 (0.9) | 24.0 (2.6) |
| 30-40 | n.d. | 50 (4) | 10.7 (0.3) | 0 | 236 (21) | 225 (22) | 15.8 (1.7) | 20.6 (1.1) | 27.6 (2.3) |
| **Wetzstein** | | | | | | | | | |
| 0-10 | 3.5 (0.0) | 76 (15) | 24.6 (2.2) | 0 | 264 (164) | 250 (165) | 9.2 (6.2) | 17.0 (10.4) | 1.5 (0.8) |
| 10-30 | 3.8 (0.3) | 60 (29) | 22.0 (3.6) | 0 | 219 (40) | 344 (123) | 24.9 (15.5) | 37.0 (12.6) | 4.4 (0.8) |
| 30-50 | 4.2 (0.3) | 45 (24) | 19.1 (2.3) | 0 | 221 (46) | 364 (63) | 17.4 (16.6) | 27.4 (16.1) | 7.8 (2.6) |
| **Gebesee** | | | | | | | | | |
| 0-5 | 6.8 (0.1) | 26 (2) | 11.3 (0.9) | 0 | 28 (5) | 345 (5) | 1.4 (0.1) | 6.9 (0.3) | 1.3 (0.1) |
| 10-20 | 7.0 (0.4) | 22 (2) | 10.5 (0.1) | 0 | 26 (3) | 336 (2) | 1.4 (0.2) | 6.9 (0.3) | 1.4 (0.1) |
| 30-40 | n.d. | 17 (1) | 11.1 (0.2) | 1.6 (0.9) | 21 (1) | 368 (1) | n.d. | n.d. | n.d. |





**Table 2: Distribution of C species in organic matter extracted into 0.8 M NaF–0.2 M NaOH from heavy fractions of mineral topsoil**
**(0–5 cm depth) and subsoil (30–40 or 30–50 cm depth) layers as revealed CPMAS-$^{13}$C-NMR.**

| Sample | Carbonyl/carboxyl C 220–160 ppm | Phenolic/aromatic C 110–160 ppm | O/N-alkyl C 45–110 ppm | Alkyl C –10–45 ppm |
|---|---|---|---|---|
| | | | % | |
| Hainich 0–5 cm | 6 | 8 | 63 | 24 |
| Hainich 30–40 cm | 5 | 3 | 74 | 18 |
| Hesse 0–5 cm | 8 | 6 | 62 | 24 |
| Hesse 30–40 cm | 5 | 3 | 67 | 25 |
| Laqueuille 0–5 cm | 9 | 8 | 62 | 21 |
| Laqueuille 30–40 cm | 9 | 6 | 63 | 22 |
| Wetzstein 0–10 cm | 14 | 17 | 35 | 34 |
| Wetzstein 30–50 cm | 14 | 9 | 47 | 30 |
| Gebesee 0–5 cm | 9 | 14 | 52 | 25 |
| Gebesee 30–40 cm | 6 | 15 | 60 | 19 |





**Table 3: Distribution of C species in total, 0.8 M NaF–0.2 M NaOH-extractable and residual organic matter of the heavy fraction of the mineral topsoil layer (0–5 cm depth) at site Hainich as revealed CPMAS-[13]C-NMR.**

| Sample | Carbonyl/carboxyl C 220–160 ppm | Phenolic/aromatic C 110–160 ppm | O/N-alkyl C 45–110 ppm | Alkyl C −10–45 ppm |
|---|---|---|---|---|
| | ——————— % ——————— | | | |
| Total OM | 8 | 14 | 52 | 26 |
| Extracted OM | 6 | 8 | 63 | 24 |
| Residual OM | 9 | 15 | 50 | 26 |






Appendix A: Supplementary Figures.

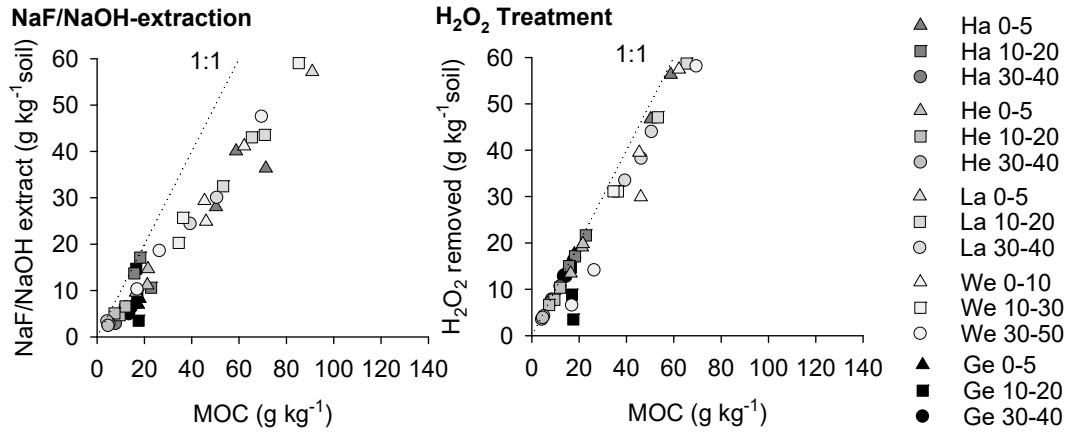


**Figure A1: Dependency of removed OC from original concentrations of mineral-associated OC for the NaF/NaOH extraction (left) or the $H_2O_2$ oxidation (right).**





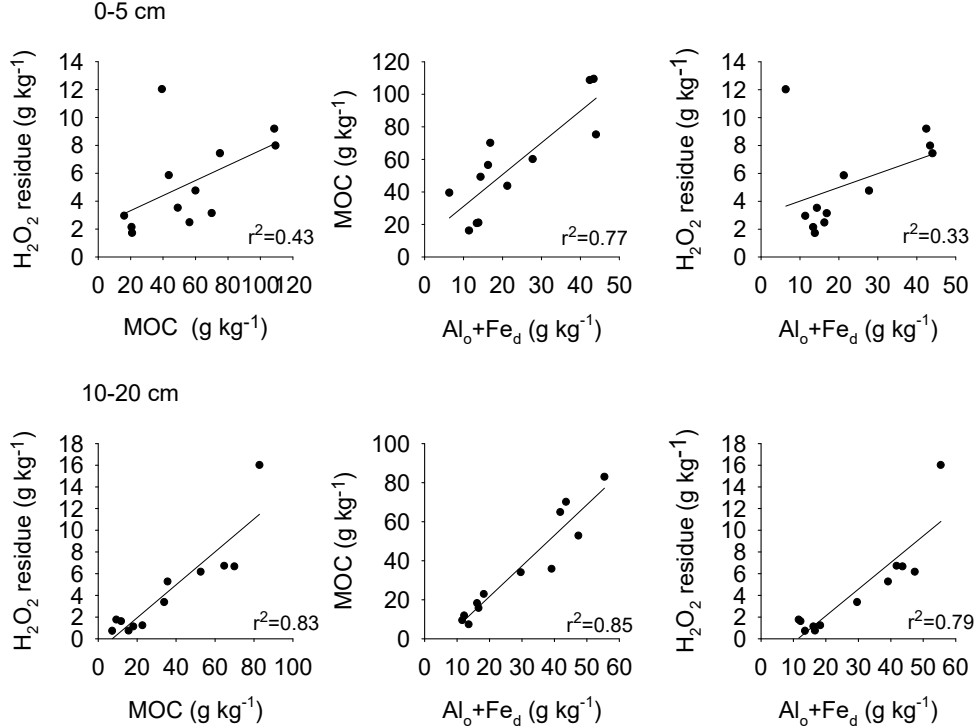


**Figure A2: Relations between OC in mineral association (MOC), the amount of OC left from MOC after the H₂O₂ treatment (H₂O₂ residue) and the content of oxalate extractable Al (Al_o) and dithionite extractable Fe (Fe_d) for two soil depths (Al_o and Fe_d data were taken from Schrumpf et al. (2013)).**






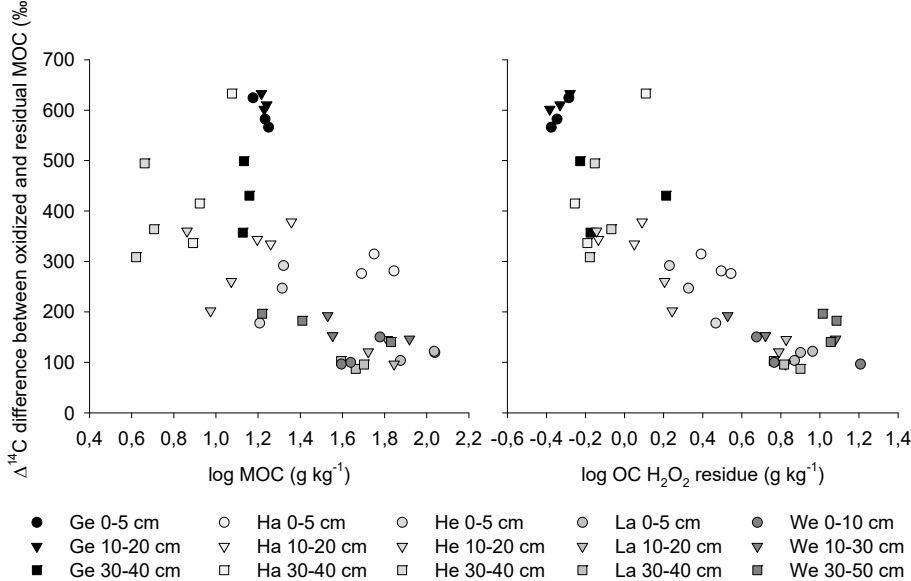

**Figure A3: Decline in $^{14}$C differences between oxidized and residual OC after $H_2O_2$ treatment with increasing OC amounts in total MOC and OC left in $H_2O_2$ residues across sites and depths.**






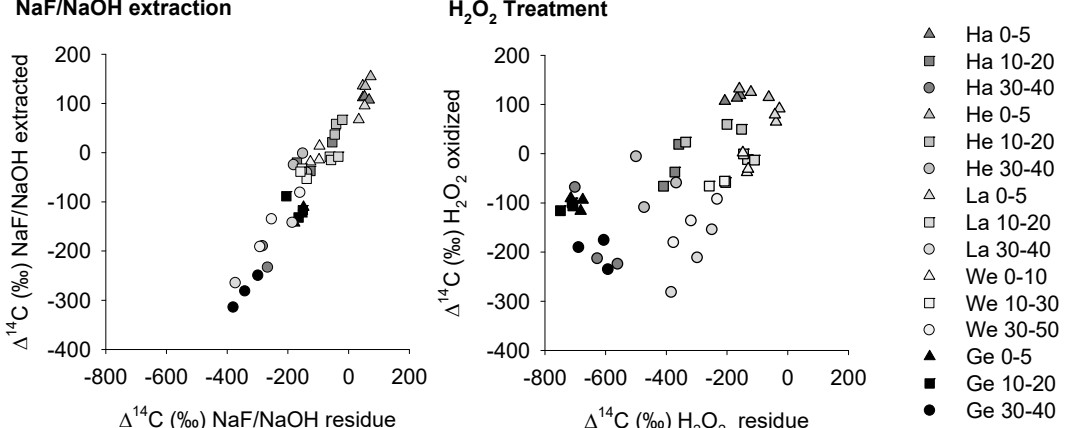

**Figure A4: Correlation between ¹⁴C contents of NaF/NaOH extracts and extraction residues of HF-OC (left) and between oxidized OC and residue OC following treatment of HF-OC with heated H₂O₂ (right).**