# Peer review of "Age distribution, extractability, and stability of mineral-bound organic carbon in central European soils"

_Biogeosciences, 2020_

## Referee Comment (RC1) · Anonymous Referee #1 · 28 Jul 2020

The manuscript reports on experiments characterizing mineral associate organic matter using a comparison of "extraction" by NaOH-NaF versus "oxidation" by H2O2. The treatments were coupled with 13C-NMR and 14C dating. The experiments are well designed according to current paradigms of SOM stabilization, but the manuscript requires substantial revision before it might be acceptable for publication.

There are a large number of instances throughout the manuscript where diction and grammar are awkward or incorrect. The manuscript should be deeply reviewed by a native English writer to revise these subtleties.

The end of the introduction should be restructured to move ln96-104 to the end of the

section, where it will serve as a segue to the methods section. That is, I recommend an introduction with the structure: background, problem statement, hypotheses, ways to test them.

The hypotheses listed are not hypotheses sensu stricto because they are not testable in the strictest way. That is, they cannot be answer with a simple "yes" or "no". Several hypotheses have multiple conditions or clauses that should be broken into several subhypotheses. I am not necessarily a purist when it comes to these formulations, but when I know this, I typically replace the word "hypothesis", which can be reserved for strict statistical or logical uses, with the word "expectation". All this being said, I do recommend trying to break up the 5 bullets into expectations and outcomes. That is, it appears that some of these so-called hypotheses are actually expectations, which if are true, other conditions would also be true. Others are more contradictory, where if not true then... A clearer, more explicit and deliberate structure for all these expectation will result in better structured results and discussion sections.

A few minor points in the methods section: ln141: filters are normally described at least by their pore-size and sometimes their diameter, not their diameter only. ln161-162: The sentence on carbonates is unnecessary and likely the result of copy-paste text because ln158-159 already state that all soils were carbonate-free.

In some instances the order in which results are circuitous and confusing. Perhaps use the "hypotheses" or methods section as road maps for ordering the presentation of results.

I like how the discussion section is structured. It places the results in a clear context. However, I'm not convinced that the results are "surprising", and I find the overall interpretation to be a little off the mark. I did not find it surprising that the proportions (vs. "portions" which is incorrectly used throughout the manuscript) were consistent across most soils. This has been previously been observed for both H2O2 (eg, Plante 2014 EJSS) and acid hydrolysis (eg, Paul 2006 SSSAJ). This is one of the problems with

chemical extractions: they rarely demonstrate the expected trends in inferred stability. A similar lack in expected trends has also been frequently observed in 14C dates. So, given that a substantial proportion of the results from this study did not meet expectations, I would strongly recommend reframing the manuscript. It might be much more compelling to more specifically spell out what the conceptual framework (paradigm) is that leads to the expectations outlined. The goal of the discussion would then be to describe where the problems are with either the assumptions etc. in the conceptual framework, or in the methods used to test them. The manuscript currently tries to address the former, but not necessarily the latter. Are NaOH-NaF and H2O2 appropriate tools for probing SOM stability? OR is our conception of SOM stability incorrect? The match between expectations and results that I am referring to is well illustrated in the diction of the subheader in ln344. The use of the word "Missing" suggests it was expected a priori. A more objective and unbiased approach would be to refer to it is "lack of".

I also found it unsurprising that the chemistry of extracted OC was similar across samples and dominated by polar molecules (eg, alkyl). In essence, the experiment demonstrated the solubility of a polar fraction of OM in a highly polar solution. It would not be a reasonable expectation to see non-polar OM (eg, aryl) in such a polar solution, or vice-versa.

The tables and figures used to report the results are appropriate, though the figures are numerous.

While it is appropriate to report the 1:1 line in Fig7 (modeled vs. measured), I'm not sure why it is reported in Fig3. It seems to me the slope represents the proportion extractable. If I had to guess, the slope would be 0.58 (or its inverse). I'm not sure what it would mean to have the data fall on the 1:1 line in this case.

I have become increasingly frustrated by the visual and qualitative interpretation of spectral data using a stack of "squiggly lines". While large differences might easily

be apparent, smaller, more subtle differences, or large differences in smaller peaks may not be so apparent. Differences among spectra should be tested quantitatively/statistically, perhaps using a multivariate method such as PCA or NMDS.

---

## Referee Comment (RC2) · Anonymous Referee #2 · 11 Aug 2020

Overall, I thoroughly enjoyed this manuscript. However, I am concerned that much of the language in the discussion is speculative in nature. I do not believe that reasonable speculation should not be allowed, as true statistical replication in soil studies is sometimes difficult to achieve due to measurement cost (NMR) or sample availability. But the number of statements basing interpretation of mechanism or process which are based on data from one sample or site seems very high in this manuscript. The authors could focus solely on interpretation of how extraction chemistry influences the portion and composition of extractable organic matter, and that would be sufficient.

The comments linking NaF-extractable C with Al/Fe oxyhydroxides don't seem well

supported by the data. Figure A2 illustrates a significant relationship between H2O2 residues and extractable Al/Fe. Why are there not similar plots for NaF residues?

From the abstract it isn't clear from the abstract why these two particular extractants were chosen. This may be a major gap in my knowledge base, and if that is the case, please ignore. I have never previously seen an experiment that utilized a mixture of NaOH and NaF to evaluate SOM extractability/solubility. Is there a reference that can be associated with this method? If the abstract included an explanatory statement, it would be very helpful. It sounds as if H2O2 is meant to represent oxidizable SOM, which the authors may be arguing is not very representative of the portion of SOM which is readily exchangeable under field conditions. In comparison, a NaOH and NaF solution presumably extracts SOM which is actually bound on the exchange surfaces of minerals. If this is indeed the case, it needs to be stated more explicitly in the abstract. Maybe something similar to the statement on lines 102-104 of the introduction.

In the abstract, the language describing trends in radiocarbon data uses all the following terms: 14C, 14C content, MO14C, 14C-depleted, older, and age. It would improve the clarity of the results if the language was more consistent when describing this data. Since depletion and enrichment are relative terms (depleted in comparison to what?), it seems necessary to include actual values with associated error terms when describing significant differences between extractions and residues.

The abstract fails to mention that the soils were density fractionated prior to conducting the extractions.

It is unclear in the abstract how the experimental results could lead to the conclusion that MOC was dominated by OC interactions with pedogenic oxides. This seems like a complete non sequitur.

Throughout the manuscript, it would increase the clarity of the arguments and results if the language regarding radiocarbon analysis was cleaned up. I hate to argue semantics, but I believe the terms 'stability' and 'lability' are now often rejected by the

community due to lack of specificity. Colleagues have been insisting on using the term 'persistence' as far as I know. I also believe 'older', 'younger', and 'age' are not appropriate for use in a manuscript, unless in reference to a 'mean system age'.

In hypothesis #5, it is unclear what the term 'organic acids' is referring to.

In the figures, it would be helpful to readers if the soils were referred to according to their dominant physicochemical attribute rather than the site name. Hesse = loess/Cambisol; Laqueuille = grassland/Andisol... or something similar. that way the reader can immediately draw inferences based on differences in soil physicochemical characteristics. Again in section 3.3, using site names gives little information, and forces the reader to continually check back to table 1 for context.

What was the pH of the NaF/NaOH solution (13-14, correct?), and what influence do the authors think this had on the amount and characteristics of the extracted C? Perhaps the effect of using such a strongly basic solution masked any influence of F- on the amount of exchangeable C displaced on mineral surfaces.

Butnor et al., 2017 (doi: 10.1016/j.foreco.2017.01.014) is an additional reference that explores $H_2O_2$ residues and Fe oxyhydroxides.

It would offer additional insight if the authors cold measure $\delta 13C$ and $\delta 15N$ for the extracts and residues to see if there were significant differences in the degree of microbial processing. Perhaps it would be acceptable to use the 13C values from the AMS measurements? I know these aren't typically considered publishable due to possible fractionation effects, but it seems as though they could be used for intercomparison purposes if it can be assumed that all samples experienced similar degrees of fractionation during AMS measurement?

Additional NMR data would also be interesting. Would it be possible to measure the $H_2O_2$ residuals?

---

## Author Comment (AC3) · 19 Sep 2020

Supplement:

1) As recommended by referee 2, we attempted gaining additional evidence on the origin of NaF-NaOH-extractable and residual OC using the $^{13}C$ data from the AMS of the $^{14}C$-laboratory. However, as shown below for the tree forest sites (average of three samples per depth and standard deviation), extracts were neither consistently enriched nor depleted in $^{13}C$, so that no general conclusion on, e.g., differences in the degree of microbial processing can be drawn. Given the uncertainties of AMS-based $^{13}C$ data, we would prefer not to include them in the manuscript.

[Figure]

2) In order to enable better distinction between the original expectations regarding the effect of desorption on OC persistence and $^{14}C$ contents, and the new hypotheses derived from the unexpected results, we suggest adding the following graphical summary to the conclusions of the revised version.

**Original expectations:**

[Figure]

**Observations and new hypotheses:**

[Figure]

**E1**: Persistence of MOC is due to resistance to desorption in NaF/NaOH

**E2**: OC resistance to desorption is affected by:
- mineral composition
- OC loading of minerals
- vegetation type

For acid to neutral forest and grassland soils:

**H1:** Similar extractability is due to similar dominant bond types

**H2:** Differences in $^{14}C$ are mostly due to input and exchange with DOC

---

## Author Response (AR1)

**Dear editor,**

We are glad that you are considering our manuscript for publication after revision. Therefore, we implemented the answers already prepared and submitted for the two referees before in a now substantially revised version of the manuscript. Below you will find the referee comments and our answers again, as well as the marked-up version of the new manuscript.

5  Kind regards,

Marion Schrumpf

(on behalf of the authors)

We would like to thank the referee for taking the time to read and comment on our manuscript, which will help us
10  improve its clarity in a revised version.

Answers to individual points raised follow below.

**Referee 1**

The manuscript reports on experiments characterizing mineral associate organic matter using a comparison of "extraction" by NaOH-NaF versus "oxidation" by H2O2. The treatments were coupled with 13C-NMR and 14C dating. The experiments
15  are well designed according to current paradigms of SOM stabilization, but the manuscript requires substantial revision before it might be acceptable for publication.

A revised manuscript will be prepared based on the recommendations an suggestions provided by the two referees.

20  There are a large number of instances throughout the manuscript where diction and grammar are awkward or incorrect. The manuscript should be deeply reviewed by a native English writer to revise these subtleties.

As recommended, we will adopt the semantic changes suggested by the two referees, and the two native English speaking co-authors of the manuscript will double-check the overall language again for awkward wording and syntax
25  flaws.

The end of the introduction should be restructured to move ln96-104 to the end of the section, where it will serve as a segue to the methods section. That is, I recommend an introduction with the structure: background, problem statement, hypotheses, ways                                                                to                                                     test                                                           them.

We will transfer the description of the methods used to test the hypotheses to the end of the introduction to obtain the structure recommended by the referee.

The hypotheses listed are not hypotheses sensu stricto because they are not testable in the strictest way. That is, they cannot
35  be answer with a simple "yes" or "no". Several hypotheses have multiple conditions or clauses that should be broken into several subhypotheses. I am not necessarily a purist when it comes to these formulations, but when I know this, I typically replace the word "hypothesis", which can be reserved for strict statistical or logical uses, with the word "expectation". All this being said, I do recommend trying to break up the 5 bullets into expectations and outcomes. That is, it appears that some of these so-called hypotheses are actually expectations, which if are true, other conditions would also be true. Others are
40  more contradictory, where if not true then... A clearer, more explicit and deliberate structure for all these expectation will result in better structured results and discussion sections.

We agree having several conditions in one hypothesis hampers simple yes/no answers to whether or not a hypothesis is supported by the data, as the answer could depend on the conditions.

We suggest rephrasing and simplifying the "hypothesis" in the following way:

Based on the literature review we expect that:

    (1) extraction in NaF/NaOH releases a weaker bound fraction of total MOC, which is younger than the stronger bound, probably better stabilized residue fraction.
    (2) the proportion of NaF/NaOH-extractable MOC decreases with increasing contents of pedogenic oxides, which form strong bonds with OC.
    (3) the proportion of NaF/NaOH-extractable MOC declines from topsoils to subsoils due to declining OC loading of minerals.
    (4) MOC of soil samples with higher portions of extractable carbon should be younger than MOC with more extraction resistant OC.
    (5) the chemical composition of extractable MOC varies between study sites due to differences in vegetation composition, and thus litter chemistry.
    (6) the chemical composition of extractable MOC changes with soil depths due to declining contributions of plant and increasing contributions of microbial derived OC.
    (7) MOC should be less prone to desorption and accordingly of older $^{14}$C age when organic molecules forming MOC have many carboxyl groups that enable strong bonds with minerals surfaces.
    (8) the strong oxidizing agent $H_2O_2$ removes more OC from MOC than NaF/NaOH.
    (9) oxidizable and non-oxidizable OC should both be older than the extractable and non-extractable OC fractions if the harsher oxidation treatment removes more, and thus presumably better stabilized OC from mineral surfaces, leaving OC residues of even older $^{14}$C age behind.

A few minor points in the methods section:
ln141: filters are normally described at least by their pore-size and sometimes their diameter, not their diameter only.

The diameter was 1.6 µm and this information will be added to the revised version.

ln161-162: The sentence on carbonates is unnecessary and likely the result of copy-paste text because ln158-159 already state that all soils were carbonate-free.

The referee is right that carbonate-free samples were chosen, and the sentence will be deleted in the revised version.

In some instances the order in which results are circuitous and confusing. Perhaps use the "hypotheses" or methods section as road maps for ordering the presentation of results.

We assume that mixing of ideas on the effects of NaF-NaOH and $H_2O_2$ was possibly confusing. Therefore we suggest to restructure the results section as follows: First, presentation of the extraction efficiency of NaF-NaOH, followed by the chemical composition of the extracts (NMR), and then of the $^{14}$C data. Thereafter, presentation of the oxidation efficiency and $^{14}$C contents of $H_2O_2$ treated samples. Finally, a comparison of results of the two treatments will be presented.

I like how the discussion section is structured. It places the results in a clear context. However, I'm not convinced that the results are "surprising", and I find the overall interpretation to be a little off the mark. I did not find it surprising that the proportions (vs."portions" which is incorrectly used throughout the manuscript) were consistent across most soils. This has been previously been observed for both H2O2 (eg, Plante 2014 EJSS) and acid hydrolysis (eg, Paul 2006 SSSAJ). This is one of the problems with chemical extractions: they rarely demonstrate the expected trends in inferred stability. A similar lack in expected trends has also been frequently observed in 14C dates.

The surprising result was not the lack of a trend in $H_2O_2$ oxidation but the uniform extractability in NaF-NaOH.
While the two references mentioned by the referee (Plante et al. 2004?) indeed found similar proportions of soil OC removed by $H_2O_2$ treatment and acid hydrolyses, others did not. For example, Eusterhues et al. (2005 Organic Geochemistry) showed a strong depth dependence of $H_2O_2$- or $Na_2S_2O_8$-oxidizable proportions of bulk OC (between 5 and 58% of OC resisted oxidation with $H_2O_2$) and also Kleber et al (2005 EJSS) found a range of 28-87% for oxidation-resistant OC across different forest subsoils. After all, while studies on OC oxidation and hydrolysis were frequently done before and address rather the inherent chemical stability of soil OM, the focus of our study was on using OC extraction into NaF-NaOH as an indicator of the stability of the bonding between OC molecules and mineral surfaces (which is also why mineral associated and not bulk soil OC was studied). The controls on OC mobilization are therefore not necessarily the same and can be expected to be closer linked to mineral composition and OC loading of minerals, as indicated by previous results from the literature. The same applies to $^{14}C$ data. Therefore, the focus of the discussion was also on the NaF-NaOH results, while little emphasis was placed on the results of oxidation resistance since not specifically surprising.

We will replace "portion" by "proportion" in the revised version of the manuscript.

So, given that a substantial proportion of the results from this study did not meet expectations, I would strongly recommend reframing the manuscript. It might be much more compelling to more specifically spell out what the conceptual framework (paradigm) is that leads to the expectations outlined. The goal of the discussion would then be to describe where the problems are with either the assumptions etc. in the conceptual framework, or in the methods used to test them. The manuscript currently tries to address the former, but not necessarily the latter. Are NaOH-NaF and H2O2 appropriate tools for probing SOM stability? OR is our conception of SOM stability incorrect?

As the referee pointed out, we had a certain concept about MOC formation, its stability and potential drivers in mind when we started the experiment as outlined in the "expectations". Many of them were not supported by the data, which can indeed be due to a combination of wrong methods used or because our conceptual understanding needs further refinement. In any case, changing the expectations a postori to make them meet the results is no reasonable way to achieve scientific progress. Instead, the discussion should be used to identify where our conceptual understanding is not in agreement with the results, derive hypotheses for the observed mismatch including potential limitations of the experimental approach, and eventually draw a modified version of the original concept and come up with ideas for further studies. Therefore one could also argue that it is useful if an expectation is not met, because it means that we can learn something from the study that we did not know before.

Nevertheless, one result of our study was that a large portion of MOC could be more homogenous in $^{14}C$ contents than expected, while only a very small portion has very old ages. Accordingly, different chemical fractionation schemes always remove OC from the same continuum, leaving increasingly old OC behind. Therefore, chemical extractions seem to be not able to isolate distinct homogenous fractions, and other approaches are needed to explain average $^{14}C$ values and age distributions of MOC as pointed out in the conclusions.

The match between expectations and results that I am referring to is well illustrated in the diction of the subheader in ln344. The use of the word "Missing" suggests it was expected a priori. A more objective and unbiased approach would be to refer to it is "lack of".

A negative statement in the headline will probably always hint at an unexpected result. Nevertheless, we will replace "missing" by "no", which is more neutral than "lack of", which, at least to our understanding, still indicates that something was expected to be there.

I also found it unsurprising that the chemistry of extracted OC was similar across samples and dominated by polar molecules (eg, alkyl). In essence, the experiment demonstrated the solubility of a polar fraction of OM in a highly polar solution. It would not be a reasonable expectation to see non-polar OM (eg, aryl) in such a polar solution, or vice-versa.

**We agree that we were extracting the samples in a polar (aqueous) solution and that accordingly most of the extracted molecules are dissolvable in water – which is indeed not surprising given that they were probably transported and transferred to mineral surfaces via the aqueous soil solution. In general, alkyl-C and aromatic C (aryl-C) moieties would be non-polar, while carboxyl groups are polar. In that respect, we could state that it is rather surprising that the extracted fraction was dominated by alkyl-C. However, whether or not a molecule is soluble in water– similar to aryl-C- dependents on the type and number of functional groups attached. It is not possible to infer the solubility or polarity of extracted molecules from their NMR spectra, as these give only information on bond types of elements but not molecules. Therefore, we think that the use a polar NaF-NaOH extraction solution was not the reason for the observed results.**

The tables and figures used to report the results are appropriate, though the figures are numerous.

While it is appropriate to report the 1:1 line in Fig7 (modeled vs. measured), I'm not sure why it is reported in Fig3. It seems to me the slope represents the proportion extractable. If I had to guess, the slope would be 0.58 (or its inverse). I'm not sure what it would mean to have the data fall on the 1:1 line in this case.

**We agree that a visualization of the different efficiency of the two treatments in removing MOC does not need visualization with the 1:1 line. We will remove it.**

I have become increasingly frustrated by the visual and qualitative interpretation of spectral data using a stack of "squiggly lines". While large differences might easily be apparent, smaller, more subtle differences, or large differences in smaller peaks may not be so apparent. Differences among spectra should be tested quantitatively/statistically, perhaps using a multivariate method such as PCA or NMDS.

**We agree the simple visual interpretation of the NMR-data from figures is largely qualitative. Therefore the figures were complemented by proper quantification of the relative contribution of C species (see Methods) to total OC in Tables 2 and 3. Since the overall result is that differences were small, scaling them into a PCA or NMDS would possibly lead to an over-interpretation of small differences probably not relevant for explaining differences in desorption or $^{14}$C ages, and thus for the topic of this study. Therefore, we do not think that adding multivariate statistics would add further information to our conceptual understanding of MOC stability.**

**We would like to thank the referee for taking the time to read and comment on our manuscript, which will help us improve the abstract and the overall manuscript.**

**Answers to individual points raised can be found below.**

**Referee 2**

Overall, I thoroughly enjoyed this manuscript. However, I am concerned that much of the language in the discussion is speculative in nature.

**We agree with the referee that the discussion contains some speculation. The reason is that many results did not meet the expectations and that we therefore offer alternative hypotheses to explain them. We will make this point clearer and reduce the degree of speculation in the revised version.**

I do not believe that reasonable speculation should not be allowed, as true statistical replication in soil studies is sometimes difficult to achieve due to measurement cost (NMR) or sample availability. But the number of statements basing interpretation of mechanism or process which are based on data from one sample or site seems very high in this manuscript.

**We agree with the referee that more real replicates would help to underline the obtained results and others might prove us wrong in the future. However, for now we have to interpret the pattern that we see in the obtained result. With five study sites from different places in Europe and true replicates within the sites we are also not at the lower end of sample numbers being analysed in experimental studies considering topsoils and subsoils. The main result of our study is anyhow that the sites have more in common than we expected and so we focused the discussion rather on common patterns than on differences among individual sites. We agree that with only one sample being analysed for the NMR results of bulk MOC and MOC residues, the observed differences can only hint at potentially underlying mechanisms and mostly serve a basis for developing new hypothesis to be further tested in the future. The similarity of the spectra of extracted OC among sites and the uniform extractability suggest, however, that the idea of similar trends at other sites is not fully unrealistic. Nevertheless, we will reduce the degree of speculation and highlight the new hypotheses more clearly in the revised version. Besides the costs, also technical problems with low signal-to-noise ratios in C-poor and oxide-rich samples reduced the broader application of NMR for more samples in our study.**

The authors could focus solely on interpretation of how extraction chemistry influences the portion and composition of extractable organic matter, and that would be sufficient. The comments linking NaF-extractable C with Al/Fe oxyhydroxides don't seem well supported by the data. Figure A2 illustrates a significant relationship between $H_2O_2$ residues and extractable Al/Fe. Why are there not similar plots for NaF residues?

**Since we already showed in Schrumpf et al. (2013) that HF-OC was closely related to Al/Fe oxyhydroxides and NaF/NaOH extractable OC represented roughly 60% of the total HF-OC across sites, we thought it would not be necessary to show that plot. However, we are happily willing to add the graph shown below.**

[Figure]

From the abstract it isn't clear from the abstract why these two particular extractants were chosen. This may be a major gap in my knowledge base, and if that is the case, please ignore. I have never previously seen an experiment that utilized a mixture of NaOH and NaF to evaluate SOM extractability/solubility. Is there a reference that can be associated with this method? If the abstract included an explanatory statement, it would be very helpful.

First of all, we thank the referee for the critical look at the abstract and are happy to extend the description and motivation of the methods further if the editor agrees that such an extension is not too detailed for an abstract and would make it overly lengthy.

We assume that the referee refers to the combination of NaF and NaOH as extractants since we thought that the reason for using NaF/NaOH and $H_2O_2$ was explained in the abstract. Both, NaOH alone and in combination with NaF have been used and published in the literature as extractants for OC before (see e.g. Möller et al. 2000 Aust. J. Soil Res., Kaiser et al. 2007 SSSAJ). The reasons for using NaOH were that it has the power to remove much of previously sorbed dissolved organic carbon from mineral surfaces, and because the high pH increases the deprotonation, and thus, the solubility of organic acids. Since NaF is more strongly competing for binding sites while less dispersive than $Na_4P_2O_7$, the combination of NaF-NaOH was used to study a potential maximal desorption. However, since it is not common to have references in the abstract, we would keep those in the main manuscript, but add the following sentence explaining the reason for using this extract in the abstract:
"The combination of NaF-NaOH was used because F⁻ is a strongly sorbing anion capable to replace anionic organic molecules from mineral surfaces and the high pH of the extract additionally supports desorption and solubility of MOC."

It sounds as if H2O2 is meant to represent oxidizable SOM, which the authors may be arguing is not very representative of the portion of SOM which is readily exchangeable under field conditions. In comparison, a NaOH and NaF solution presumably extracts SOM which is actually bound on the exchange surfaces of minerals. If this is indeed the case, it needs to be stated more explicitly in the abstract. Maybe something similar to the statement on lines 102-104 of the introduction.

Actually, we already wrote in lines 13-15 in the abstract: "Therefore, we determined the extractability of MOC into a mixture of 0.1 M NaOH and 0.4 M NaF as a measure for maximal potential desorbability, and compared it with maximal potential oxidation in heated $H_2O_2$." We believe that this sentence already demonstrates the main intention for using the two different treatments. It can be complemented by stating that they address different aspects of MOC stability: $H_2O_2$ addresses the chemical stability of the molecules, and NaF-NaOH the possible displacement and mobilization of the molecules from mineral surfaces by competing ions.

In the abstract, the language describing trends in radiocarbon data uses all the following terms: 14C, 14C content, MO14C, 14C-depleted, older, and age. It would improve the clarity of the results if the language was more consistent when describing this data.
Since depletion and enrichment are relative terms (depleted in comparison to what?), it seems necessary to include actual values with associated error terms when describing significant differences between extractions and residues.

Following the recommendation of the referee, we will add absolute [14]C values and error measures to the abstract and harmonize the language used to describe the results.

The abstract fails to mention that the soils were density fractionated prior to conducting the extractions.
To fulfil this request, we will change the sentence in lines 15ff. in the following way:
"We selected MOC samples (>1.6 g cm³) obtained from density fractionation of samples from three soil depth increments (0-5 cm, 10-20 cm, 30-40 cm) of five typical soils of the mid-latitudes, differing in contents of clay and pedogenic oxides, and being under different land use."

It is unclear in the abstract how the experimental results could lead to the conclusion that MOC was dominated by OC interactions with pedogenic oxides. This seems like a complete non sequitur.

The conclusion on the role of pedogenic oxides in lines 36ff of the abstract is just repeating what was stated and introduced before. We would kindly like to draw the attention of the referee to lines 16 and lines 22-23 of the abstract. In line 16, we mentioned that sites were chosen to have some spread of pedogenic oxide contents. In lines 22-

**23, we stated: "Total MOC amounts were linked to the content of pedogenic oxides across sites, independent of variations in total clay. The uniform MOC desorption could therefore be the result of pedogenic oxides dominating the overall response of MOC to extraction." We can make this point more clear by adding that also MOC residues were correlated to pedogenic oxides. Therefore, one explanation for the uniform extractability of MOC is that MOC is dominated by interactions between OC and pedogenic oxides and that clay minerals were less important.**

Throughout the manuscript, it would increase the clarity of the arguments and results if the language regarding radiocarbon analysis was cleaned up. I hate to argue semantics, but I believe the terms 'stability' and 'lability' are now often rejected by the community due to lack of specificity. Colleagues have been insisting on using the term 'persistence' as far as I know. I also believe 'older', 'younger', and 'age' are not appropriate for use in a manuscript, unless in reference to a 'mean system age'.

**We are not convinced that there is consensus in the community when it comes to terminology and on whether or when to use the term persistence instead of stability. The term "persistence" only became popular after its usage in the title of the Schmidt et al. (2011) paper and therefore stands maybe for replacing the old chemistry-based "stability" (=recalcitrance?)-paradigm by a new one. In that sense, using persistence in our manuscript might be appropriate but the term has not been clearly defined in the Schmidt et al. paper. We think it is reasonable to state that OC can become temporarily stabilized, e.g. by adsorption to mineral surfaces, and that it might be re-mobilized later on. Also, we think it makes more sense to directly refer to "stabilization processes" instead of "processes increasing the persistence of OC soils". Both terms might have their justification and will re-check the manuscript for their correct use.**
**Regarding the $^{14}$C results, we did not calculate any absolute ages, mean ages, age distributions or transit times of OC in samples, since this exercise requires models with specific assumptions. Irrespective of the reference system, it is correct to state that carbon in samples with smaller $^{14}$C contents or more negative $\Delta^{14}$C values than in other samples had been isolated from the atmosphere for a longer time and can accordingly be referred to as being older. We will add a respective sentence to the methods part to make clear what we mean when referring to younger or older carbon.**

In hypothesis #5, it is unclear what the term 'organic acids' is referring to.

**Hypthesis 5 will be reworded in the revised version following the recommendations of referee 1 and will now read: "MOC should be less prone to desorption and accordingly of older $^{14}$C age, when organic molecules forming MOC have many carboxyl groups that enable strong bonds with minerals surfaces."**

In the figures, it would be helpful to readers if the soils were referred to according to their dominant physicochemical attribute rather than the site name. Hesse = loess/Cambisol; Laqueuille = grassland/Andisol: : or something similar. That way the reader can immediately draw inferences based on differences in soil physicochemical characteristics.

**We thank the referee for this suggestion and will complement the figure caption with a brief soil/site description.**

Again in section 3.3, using site names gives little information, and forces the reader to continually check back to table 1 for context.

**We agree and will provide the respective context in the revised version.**

What was the pH of the NaF/NaOH solution (13-14, correct?), and what influence do the authors think this had on the amount and characteristics of the extracted C? Perhaps the effect of using such a strongly basic solution masked any influence of F on the amount of exchangeable C displaced on mineral surfaces.
**The referee is right that besides F⁻ also OH⁻ anions from NaOH in the extraction solution will compete with and remove organic molecules from mineral surfaces. The higher pH will also lead to a deprotonation of organic acids.**

**Since both, NaF and NaOH affect desorption in the same way and direction (also NaF alone would lead to an increase in soil pH), their relative importance should be unimportant for the overall result.**

Butnor et al., 2017 (doi: 10.1016/j.foreco.2017.01.014) is an additional reference that explores H2O2 residues and Fe oxyhydroxides.

**We thank the referee for making us aware of that study. Using a combination of 20% $H_2O_2$ and 0.33 $\underline{M}$ $HNO_3$, Butnor et al. 2017 also found increasing oxidation resistance of OC with soil depths, and a relation between oxidation resistant OC and extractable Fe (Mehlich-3) and clay. We will add the reference to line 389 of the manuscript, where we are discussing the effect of pedogenic oxides for the oxidation resistance of OC.**

It would offer additional insight if the authors cold measure _13C and _15N for the extracts and residues to see if there were significant differences in the degree of microbial processing. Perhaps it would be acceptable to use the 13C values from the AMS measurements? I know these aren't typically considered publishable due to possible fractionation effects, but it seems as though they could be used for intercomparison purposes if it can be assumed that all samples experienced similar degrees of fractionation during AMS measurement?

**We thank the referee for this suggestion. We actually already tested the potential application of the OC-to-N ratio for a similar reason but did not obtain consistent, and thus, easily interpretable results, also because N concentrations in (subsoil) residues were sometimes very small. $^{15}N$ analyses cannot be added any more now because for many fractions the sample material was little and used up during analyses. The same applies for additional $^{13}C$ analyses. For the reason mentioned by the referee, we typically do not use the $^{13}C$ values of the AMS but will evaluate their application for the revised version and add them if appropriate. Unpublished results from a previous experiment showed that both, oxidation and NaOH extraction residues were depleted in $^{13}C$ (and $^{14}C$) relative to the original sample, indicating that resistant OC was rather less microbial processed than the extracted or oxidized fractions.**

Additional NMR data would also be interesting. Would it be possible to measure the H2O2 residuals?

**From a conceptual point of view we agree that it would be great to do that. Unfortunately, it is neither technically feasible nor reasonable. First, because of the small OC concentrations in the HF-residues, where the noise produced by the mineral background will impede the interpretation of the spectra even with long measurement times, and also because the $H_2O_2$ treatment probably also chemically altered the residues, so that it would be difficult to tell if the molecular structures observed now are no artefacts cause by the strong oxidation.**

Supplement:

1) After receiving the $^{13}C$ data from the AMS of the $^{14}C$-laboratory, we plotted them to test if further information can be gained on the origin of the extracted or residual OC from NaF-NaOH extractions as recommended by referee 2. However, as shown below for the tree forest sites (average of three samples per depth and standard deviation), extracts were neither consistently enriched nor depleted in $^{13}C$, so that no general conclusion e.g. differences in the degree of microbial processing can be drawn. Given the larger uncertainties of AMS-based $^{13}C$ measurements, we would prefer not include them in the manuscript.

[Figure]

Hainich δ¹³C (‰)  Hesse δ¹³C (‰)  Wetzstein δ¹³C (‰)

● NaF/NaOH extract    ○ residue

2) In order to enable a better distinction between the original expectations of the study regarding the role of desorption for OC persistence and 14C contents and the new hypothesis derived from the unexpected results, we suggest adding the following graphical summary to the conclusions of the revised version.

360

[Figure]

**Original expectations:**

O¹⁴C on layer silicates
residue | extract    ⊖⊕ ·O-C-R
O¹⁴C on pedogenic oxides
residue | **Me·O-C-R**
← persistent    labile →

OC persistence

E1: Persistence of MOC is due to resistance to desorption in NaF/NaOH
E2: OC resistance to desorption is affected by:
 - mineral composition
 - OC loading of minerals
 - vegetation type

[Figure]

**Observations and new hypotheses:**

DO¹⁴C
**Me·O-¹⁴C-R**
residue | extract
← persistent  labile →
DO¹⁴C
**Me·O-¹⁴C-R**

OC residence

[revised manuscript text omitted]
 and that OC in samples with more negative $\Delta^{14}C$ values than others washave been isolated longer from the atmosphere and is accordingly on average older. Positive $\Delta^{14}C$ values on the other hand indicate enrichment with bomb-derived $^{14}C$ from nuclear weapon testing in the early 1960ies, which suggests that OC from these samples is younger and has faster turnover than 300mostly fixed in the last century. years. For samples collected in the yearIn 2004, as in the present project studywhen samples in this study were collected, atmospheric $\Delta^{14}C$ values (about 70 ‰) were alreadystill declining from the peak bomb C in 1963 of about +900 ‰. Thus, comparing oxidized or extracted C with the original MOM depends on its original $\Delta^{14}C$ values. For total MOM with negative $\Delta^{14}C$ values, we assume the extracted or oxidized C is 'younger' if it has $\Delta^{14}C$ values that are either less negative or positive. For total MOM that initially has positive $\Delta^{14}C$ values, we might expect 'older' C residues to have either higher $\Delta^{14}C$ values (closer to the bomb peak) or negative $\Delta^{14}C$ values; in both cases we expect 'younger' extracted $\Delta^{14}C$ values to have positive $\Delta^{14}C$ values. Our pFor soil C pools with turnover times between 20 and 300 years, higher $\Delta^{14}C$ values however still indicate more bomb-C enrichment and thus faster turnover. Only for samples withFor samples with turnover times of less than 20 years, smaller $\Delta^{14}C$ values would indicate faster turnover and thus samples with younger carbon., but G given our previous results based on temporal changes of $\Delta^{14}C$ values of MOC at the Hainich site ({Schrumpf, 2015 #26}), it does not seemis not plausible that the largest sharemajority ofindicate that total MOC in our soils will turnover faster thanhave turnover times > have turnover times of less than 20 years. Therefore, weif extracted/oxidized OC has interpret OC in samples with morerelatively negative $\Delta^{14}C$ values that are higher than unextracted or residual MOM, than others as beingwe refer to these as younger throughout the manuscript. older, and samples with more positive values compared to othersrelatively positive values as containing on average younger carbon throughout the manuscript.

To determine the amount of carbon in the residues from both extraction procedures, we multiplied the measured OC content in the recovered residues with its mass. The amount of carbon lost by the treatments was determined as difference between

the original OC content of the HF sample and OC in the residues. The radiocarbon contents of the $H_2O_2$ residues were directly measured, and the $^{14}C$ fraction of the OC lost/extracted ($^{14}C_{extract}$) was determined by mass balance as follows:

[revised manuscript text omitted]

**Figure 4:** Relation between OC concentrations in bulk MOC, NaF–NaOH extraction residues and oxidation residues and the contents of pedogenic oxides expressed as the sum of oxlate extractable Al and acid dithionite extractable Fe. OC concentrations in each fraction (bulk MOC, NaF-/NaOH extraction residue, and oxidation residue) was related to the pedogenic oxide content of the sample, which is expressed as the sum of the acid oxalate extractable Al and acid dithionite extractable Fe.

**3.1 Radiocarbon contents of NaF/NaOH extracts and residues**

Results for directlyDirectly measured $\Delta^{14}$C values inof dialyzed NaF/NaOH extracts were overall comparable to calculated $\Delta^{14}$C values in extracts fromusing the mass balance approach (Figure 5), suggesting that there were no systematic losses of older or younger C during the extraction and subsequent dialysis procedure. The only exception were the results from Only for Gebesee, where the mass balance approach suggests that some young carbon was probably lost during dialyseis of the extracts.

[Figure]

**Figure 5: Comparison of $^{14}$C contents of NaF/NaOH extracted OC obtained using a mass balance approach and fromwith direct measurements of the extracts after dialyses (study sites: Hainich (Ha), Hesse (He), Laqueuille (La), and Wetzstein (We) on the left, Gebesee (Ge) on the right).**

Extracted fractions had consistently largerhigher Otherwise, $\Delta^{14}$C values were always larger in the extracted fractions than in the extraction residues, and $\Delta^{14}$C values decreased at all sites with soil depths at all sites (Figure 6). The $\Delta^{14}$C values of the OC extracted from the uppermost layers increased in the order Gebesee (Chernozem) < Wetzstein (0-10 cm, Podzol) < Laqueuille (Andosol) < Hainich (Cambisol) < Hesse (Luvisol) from -126‰ to 142‰. The average difference in $\Delta^{14}$C between extracted and residue OC was 79±36‰ across sites and increased in the order Gebesee (34±4‰) < Laqueuille (38±6‰) < Hainich (63±3‰) < Hesse (84±5‰) < Wetzstein (100±15‰). As indicated by the almost parallel shifts in $\Delta^{14}$C values, there was no general trend for increasing or declining differences in $^{14}$C contents with soil depths (Figure 6). Instead, $\Delta^{14}$C values of extracts and extraction residues were highly correlated ($r^2$=0.91, p<0.01, supplementary Figure S2). However, $\Delta^{14}$C values of bulk MOC, extractable or residual MOC were , however, all unrelated to total MOC or its extractability (results not shown).

[Figure]

**Figure 6: Depth profiles of radiocarbon (Δ$^{14}$C) in bulk mineral associated OC (bulk MOC), as well as in OC removed from mineral surfaces using either NaF/NaOH or H$_2$O$_2$, and in the respective OC residues remaining on mineral surfaces.**

**3.23 NMR spectroscopy of NaF/-NaOH treated samples**

The NMR spectra of extracted OM from soils of the sites Hainich, Hesse, and Laqueille were remarkably similar and dominated by signals in the O/N-alkyl C region (62-74% of total peak area across sites and depths) and the alkyl C region (18-25%), suggesting a strong contribution of carbohydrates and aliphatic compounds to the extracted OM (Figure 47, Table 2). All six spectra also reveal a distinct peak centered around 174 ppm, due to carboxyl C, which is in line with the fact that the alkaline extraction tends to release preferentially release acidic compounds (5-9%). All spectra show small signals in the aromatic regions centered around 150 ppm (phenols) and 130 ppm (non-substituted aromatic systems). All six spectra featured signals, some even well-resolved, around 56 ppm, indicating the presence of methoxyl C.

[Figure]

[Figure]

745 **Figure 47: NMR spectra of OM extracted into NaF/NaOH from the mineral associated fraction of two soil depths from the five study sites Gebesee (Chernozem, cropland), Hainich (Cambisol, beech), Hesse (Luvisol, beech), Laqueuille (Andosol, grassland), and Wetzstein (Podzol, spruce).**

**Table 2:** Distribution of C species in organic matter extracted into NaF/NaOH from heavy fractions of mineral topsoil (0–5 cm depth) and subsoil (30–40 or 30–50 cm depth) layers as revealed CPMAS-$^{13}$C-NMR.

| Sample | Carbonyl/carboxyl C 220–160 ppm | Phenolic/aromatic C 110–160 ppm | O/N-alkyl C 45–110 ppm | Alkyl C −10–45 ppm |
|---|---|---|---|---|
| | | | % | |
| Hainich (Cambisol, beech) | | | | |
| Hainich 0–5 cm | 6 | 8 | 63 | 24 |
| Hainich 30–40 cm | 5 | 3 | 74 | 18 |
| Hesse (Luvisol, beech) | | | | |
| Hesse 0–5 cm | 8 | 6 | 62 | 24 |
| Hesse 30–40 cm | 5 | 3 | 67 | 25 |
| Laqueuille (Andosol, grassland) | | | | |
| Laqueuille 0–5 cm | 9 | 8 | 62 | 21 |
| Laqueuille 30–40 cm | 9 | 6 | 63 | 22 |
| Wetzstein (Podzol, spruce) | | | | |
| Wetzstein 0–10 cm | 14 | 17 | 35 | 34 |
| Wetzstein 30–50 cm | 14 | 9 | 47 | 30 |
| Gebesee (Chernozem, cropland) | | | | |
| Gebesee 0–5 cm | 9 | 14 | 52 | 25 |
| Gebesee 30–40 cm | 6 | 15 | 60 | 19 |

750

The spectra obtained on OM extracted from the Chernozem-type soil at site Gebesee resembles those of the sites Hainich, Hesse, and Laqueille, except for that they indicate more non-substituted aromatic systems, which is in accordance consistent with findings on the occurrence occurence occurrence of pyrogenic OM in such this soil type soil. 
[revised manuscript text omitted]
 was increasing with soil depth {Butnor, 2017 #169;Eusterhues, 2005 #107}. While the MOC analyzed in this study For the MOC samples studied here, showed only this was only a small and insignificant trend in oxidation resistance with depth., we did observe Since there was also a slightly stronger The small increase in the portion of oxidation-resistant OC at depth together with the slightly stronger decline in $\Delta^{14}$C values [14]C of oxidation residues with depth relative to bulk MOC. Thus, Accordingly, , indicate that, different from desorption resistance, oxidation resistance (, i.e., chemical recalcitrance), rather than desorption resistance is likely responsible of a small portion of MOC might becomes a bit more relevant to the formation of stable MOC with soil depth. However, tThe observed increase in $\Delta^{14}$C differencesdivergence of between oxidized and residual $\Delta^{14}$C values OC with smalleras more OC is oxidized occurred across sites OC amounts left in $H_2O_2$ residues across sites (Figure A3), suggestsing that there could also be an overall trend forof increasingly older OC the with smaller theamounts of OC amount leftOC 
[revised manuscript text omitted]

---

## Author Response (AR2)

**Dear editor,**

**we are very pleased about your positive assessment of the revised version of our manuscript and happily provide the final minor revisions. Thank you very much for the careful check of our manuscript and your helpful recommendations. Please find the point to point answers below.**

**Kind regards,**
**Marion Schrumpf on behalf of the authors**

Dear Authors,

Thank you for your careful revision of the manuscript incorporating all the reviewer comments and suggestions.

I am pleased to let you know that your manuscript can be published after a final "minor" revision of the following points :

- Abstract: Please combine separate sentences into one paragraph. Though you could separate the long introduction and methods from the results and discussion, the multiple-paragraph format is unusual. I also thought you could reduce some redundant, long sentences, as illustrated below. In the current version, the length reduces the appeal of your hard work.

**We will reduce the number of paragraphs to two and agree that the abstract is pretty long right now. Therefore we are pleased to follow the good suggestion of merging some sentences and content as indicated below.**

- Line 19 "the midlatitudes": Please indicate that your samples are only from Europe, not from other midlatitudes.
**Done**

- Lines 21-21: To reduce the very long abstract and make your points clearer, I would recommend combining these sentences, like "We expected that NaF/NaOH would extract less, younger MOC…, particularly in subsoils and soils with high contents of pedogenic oxides."
**Done**

- Lines 38-41: …the Δ14C values of oxidized OC (-50±110‰) were similar to those of OC extracted with NaF/NaOH (-51±122‰), but oxidation residues (-345±227‰) were much more 14C-depleted than in the residues of NaF/NaOH extraction (-130±121‰).
**Done**

- Lines 41-42: Please rewrite the weird expression "leaving increasingly older residues behind the more OC is removed".
**Done, replaced by: Accordingly, both chemical treatments removed OC from the same continuum, and oxidation residues were older than extraction residues because more OC was removed.**

- Lines 45-47: Readers would also expect your conclusion on the employed methodological approaches ("fractionation schemes"). You wanted to "test if maximum desorption with NaF/NaOH is

a suitable indicator for the labile proportion of MOC" (line 124). Please add if you have any conclusion or recommendation regarding the extraction methods.

**We added the following sentence: Therefore, none of the applied chemical fractionation schemes was able to explain site-specific differences in $\Delta^{14}$C values.**

- Line 56 "oxi-hydroxides" or oxyhydroxides (?): Did you intend to use this term to cover all oxides, oxyhydroxides, and hydroxides? Please clarify or define this.

**The correct inclusive term would be pedogenic aluminum (Al) and iron (Fe) oxides, oxyhydroxides, and hydroxides. Since this is two long, we will refer to them as oxides as defined now in the introduction (summarily referred to as oxides).**

- Line 126 "across Europe": across central Europe?

**Changed to "central Europe"**

- Fig. 1 and Table 1: It seems inappropriate to show these published results here in the Methods section. If you want to provide extra information from the cited paper, you could move these to Supplementary Information (or to Results if you can report something new).

**We moved Figure 1 and Table 1 to the supplement.**

- Lines 135-139: Please indicate the soil classification system you followed. Is it the World Reference Base?

- Line 398: "differing" or different?

**Changed to "the different"**

- Line 403: The unique MOC composition in the podzol-type soil "is"

**Done**

- Line 405 "dissolved organic carbon": double check if DOC has been defined at its first use and consistently used throughout the manuscript.

**Now defined and checked**

- Lines 526-527: Please briefly describe how your alternative hypotheses differ from the original outcomes listed in Fig. 10.

**We added a description of the original hypothesis to make clear how the new hypotheses differ.**

- Figs. 7-8: It would be more "reader-friendly" if you indicate major spectral regions on top of the spectra.

**Done**

- Fig. 10 caption: hypotheses explaining "variations in" Δ14C values of MOC across sites and with depth

**Done**

Additionally, we competed the legend of the former Figure 3.